# DEEP THINK WITH CONFIDENCE

**Yichao Fu**[2][†][∗], **Xuewei Wang**[1], **Hao Zhang**[2], **Yuandong Tian**[1], **Jiawei Zhao**[1][†]
[1]Meta AI, [2]UCSD
[†]Equal contribution

**Project Page:** jiaweizzhao.github.io/deepconf

## ABSTRACT

Large Language Models (LLMs) have shown great potential in reasoning tasks through test-time scaling methods like self-consistency with majority voting. However, this approach often leads to diminishing returns in accuracy and high computational overhead. To address these challenges, we introduce **Deep Think with Confidence (DeepConf)**, a simple yet powerful method that enhances both reasoning efficiency and performance at test time. DeepConf leverages model-internal confidence signals to dynamically filter out low-quality reasoning traces during or after generation. It requires no additional model training or hyperparameter tuning and can be seamlessly integrated into existing serving frameworks. We evaluate DeepConf across a variety of tasks and the latest open-source models, including Qwen3 and GPT-OSS series. Notably, on challenging benchmarks such as AIME 2025, DeepConf@512 achieves up to 99.9% accuracy and reduces generated tokens by up to 84.7% compared to full parallel thinking. Our code is available at https://github.com/facebookresearch/deepconf

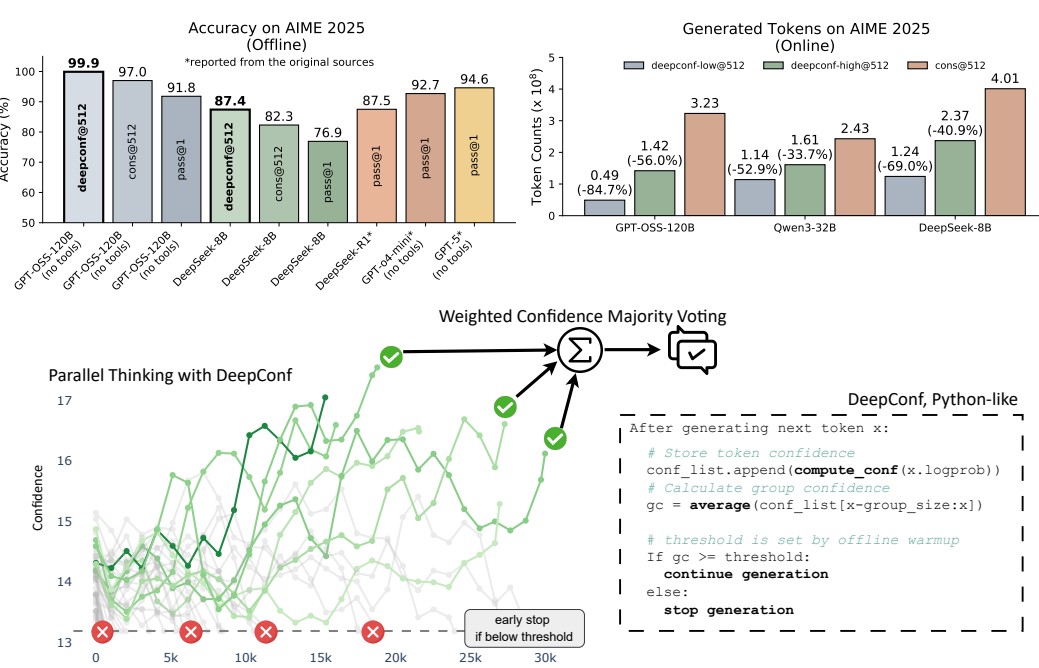

Figure 1: Up: DeepConf on AIME 2025. Down: Parallel thinking using DeepConf.

---

[∗]Work done during an internship at Meta FAIR.

## 1 INTRODUCTION

Large Language Models (LLMs) have demonstrated exceptional reasoning capabilities, particularly when equipped with methods that enhance their performance during test-time inference. A prominent technique is *self-consistency*, which samples multiple reasoning paths and aggregates final answers through majority voting (Wang et al., 2023). This type of approach, also known as *parallel thinking*, significantly improves reasoning accuracy but incurs substantial computational overhead: generating numerous reasoning traces per query scales inference overhead linearly, limiting practical deployment (Xue et al., 2023). For example, improving pass@1 accuracy from 68% to 82% using standard majority voting on AIME 2025 requires 511 additional reasoning traces per question using Qwen3-8B, consuming 100 million additional tokens.

Moreover, parallel thinking with majority voting exhibits *diminishing returns*—performance often saturates or degrades as the number of traces increase (Chen et al., 2024a). A key limitation is that standard majority voting treats all reasoning traces equally, ignoring quality variations (Pal et al., 2024; Wang et al., 2025a). This can lead to suboptimal performance when low-quality traces dominate the voting process.

Recent work has leveraged next-token distribution statistics to assess reasoning trace quality (Geng et al., 2024; Fadeeva et al., 2024; Kang et al., 2025). Higher prediction confidence typically correlates with lower entropy and reduced uncertainty. By aggregating token-level statistics such as entropy and confidence scores, existing methods compute global confidence measures across an entire trace to identify and filter low-quality traces to improve majority voting performance (Kang et al., 2025).

However, global confidence measures present several limitations in practice. First, they may obscure confidence fluctuations at local reasoning steps, which can provide sufficient signals for estimating trace quality. Averaging across entire tokens in a trace can mask critical reasoning breakdowns that occur at specific intermediate steps. Second, global confidence measures require generating complete reasoning traces before they can be calculated, which prevents early stopping of low-quality traces.

We introduce **Deep Think with Confidence (DeepConf)**, a simple yet effective test-time method that combines parallel thinking with confidence-aware filtering, based on local confidence measurements. DeepConf operates in both offline and online modes, identifying and discarding low-confidence reasoning traces either during or after generation. This approach reduces unnecessary token generation while maintaining or improving final answer accuracy.

We evaluate DeepConf across multiple reasoning benchmarks (AIME 2024/2025, HMMT 2025, BRUMO25, GPQA-Diamond) and models (DeepSeek-8B, Qwen3-8B/32B, GPT-OSS-20B/120B). Through extensive experiments averaged across 64 repetitions per setting, we demonstrate that DeepConf achieves superior reasoning performance while requiring significantly fewer generated tokens compared to standard majority voting.

In offline mode with access to all reasoning traces, DeepConf@512 achieves 99.9% accuracy on AIME 2025 using GPT-OSS-120B (no tools), saturating this benchmark compared to 97.0% for cons@512 (majority voting) and 91.8% for pass@1. In online mode with real-time generation control, DeepConf reduces token generation by up to 84.7% compared to standard parallel thinking while maintaining or exceeding accuracy. Fig. 1 highlights our key results.

## 2 CONFIDENCE AS AN INDICATOR OF REASONING QUALITY

Recent work has demonstrated that reasoning trace quality can be effectively estimated using metrics derived from the model's internal token distributions (Kang et al., 2025). These metrics provide model-intrinsic signals for distinguishing high-quality reasoning trajectories from erroneous ones without requiring external supervision.

**Token Entropy.** Given a language model's predicted token distribution $P_i$ at position $i$, the *token entropy* is defined as:

$$H_i = -\sum_j P_i(j) \log P_i(j), \tag{1}$$

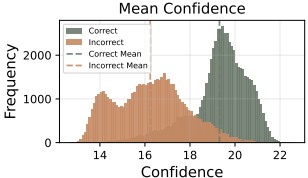 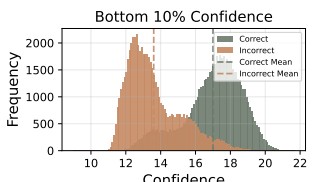 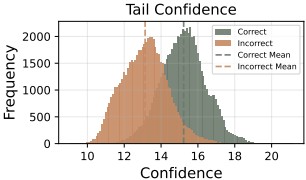

Figure 2: Confidence distributions for correct vs. incorrect reasoning traces across different metrics. Data from HMMT25: 30 problems, 4096 traces each.

where $P_i(j)$ represents the probability of the $j$-th vocabulary token. Low entropy indicates a peaked distribution with high model certainty, while high entropy reflects uncertainty in the prediction.

**Token Confidence.** We define *token confidence* $C_i$ as the negative average log-probability of the top-$k$ tokens at position $i$:

$$C_i = -\frac{1}{k} \sum_{j=1}^{k} \log P_i(j), \tag{2}$$

where $k$ denotes the number of top tokens considered. High confidence corresponds to peaked distributions and greater model certainty, while low confidence indicates uncertainty in token prediction. Note that the top-$k$ tokens used for confidence computation may differ from those used for next token sampling (specified in Table 11). We compute confidence using the top-20 token probabilities.

**Average Trace Confidence.** Token-level metrics require aggregation to assess entire reasoning traces. Following Kang et al. (2025), we employ *average trace confidence* (also termed self-certainty) as a trace-level quality measure:

$$C_{\mathrm{avg}} = \frac{1}{N} \sum_{i=1}^{N} C_i, \tag{3}$$

where $N$ is the total number of generated tokens. As demonstrated in Fig. 2, average trace confidence effectively distinguishes between correct and incorrect reasoning paths, with higher values indicating greater likelihood of correctness.

Despite its effectiveness, average trace confidence has notable limitations. First, global aggregation obscures intermediate reasoning failures: a few high-confidence tokens can mask numerous low-confidence segments, potentially hiding critical errors. Second, this approach requires complete traces for quality assessment, preventing early termination of low-quality generations and resulting in computational inefficiency.

## 3 DEEP THINK WITH CONFIDENCE

In this section, we present how to leverage confidence more effectively to improve both reasoning performance and thinking efficiency. We target two primary scenarios: *offline* and *online* thinking. Offline thinking leverages confidence to enhance reasoning performance by evaluating and aggregating information from completed reasoning traces. Online thinking incorporates confidence during token generation to improve reasoning performance and/or computational efficiency in real-time.

### 3.1 CONFIDENCE MEASUREMENTS

To address the limitations of global confidence measures like self-certainty, we introduce several alternative confidence measurements that capture local intermediate step quality and provide more fine-grained assessment of reasoning traces.

**Group Confidence.** We quantify the confidence of intermediate reasoning steps using *group confidence*. Group confidence provides a more localized and smoother signal by averaging token confidence over overlapping spans of the reasoning trace. Each token is associated with a sliding window

group $G_i$ consisting of $n$ previous tokens (e.g., $n = 1024$ or $2048$) with overlapping adjacent windows. For each group $G_i$, group confidence is defined as:

$$C_{G_i} = \frac{1}{|G_i|} \sum_{t \in G_i} C_t, \qquad (4)$$

where $|G_i|$ is the number of tokens in group $G_i$.

Estimating reasoning trace quality requires aggregating signals from group confidence. We observe that intermediate steps with *extremely low confidence* in a trace can significantly affect final solution correctness. For instance, when confidence drops sharply during reasoning with repeated low-confidence tokens like "wait", "however", and "think again", it disrupts reasoning flow and leads to subsequent errors.

**Bottom 10% Group Confidence.** To capture the effect of extremely low confidence groups, we propose *bottom 10% group confidence*, where trace confidence is determined by the mean of the bottom 10% of group confidences within the trace:

$$C_{\text{bottom-10}}(t) = \frac{1}{|G_b|} \sum_{G_j \in G_b} C_{G_j}, \qquad (5)$$

where $G_b$ is the set of groups with the lowest 10% confidence scores. Empirically, we find that 10% effectively captures the most problematic reasoning segments across different models and datasets.

**Lowest Group Confidence.** We also consider *lowest group confidence*, which represents the confidence of the least confident group within a reasoning trace—a special case of bottom 10% group confidence. This measurement estimates trace quality based solely on the lowest confidence group:

$$C_{\text{least}}(t) = \min_{G_j \in G} C_{G_j}, \qquad (6)$$

where $G$ is the set of all token groups in the reasoning trace. We discuss how lowest group confidence improves reasoning efficiency in online thinking scenarios.

**Tail Confidence.** Beyond group-based measurements, we propose *tail confidence*, which evaluates reasoning trace reliability by focusing on the final portion. This metric is motivated by observations that reasoning quality often degrades toward the end of long chains of thought, and final steps are critical for correct conclusions. In mathematical reasoning, final answer and conclusion steps are particularly important: traces that start strong but end weakly may produce incorrect results despite promising intermediate reasoning. Tail confidence $C_{\text{tail}}$ is defined as:

$$C_{\text{tail}}(t) = \frac{1}{|T_{\text{tail}}|} \sum_{t \in T_{\text{tail}}} C_t, \qquad (7)$$

where $T_{\text{tail}}$ represents a fixed number of tokens (e.g., 2048). Fig. 2 compares different confidence measurements, illustrating that both bottom 10% and tail confidence metrics better separate incorrect and correct trace distributions compared to mean confidence methods, suggesting these metrics are more effective for trace quality estimation.

## 3.2 Offline Thinking with Confidence

We now describe how to apply various confidence measurements to improve reasoning performance in offline settings. In offline thinking, reasoning traces for each problem have been generated, and the key challenge is aggregating information from multiple traces to better determine the final answer. While recent work proposes advanced methods for summarizing and analyzing reasoning traces using LLMs, we focus on standard majority voting approaches.

**Majority Voting.** In standard majority voting, the final answer from each reasoning trace contributes equally to the final decision. Let $T$ be the set of all generated traces, and for each $t \in T$, let $\text{answer}(t)$ be the answer string extracted from trace $t$. The vote count for each candidate answer $a$ is:

$$V(a) = \sum_{t \in T} I(\text{answer}(t) = a),$$

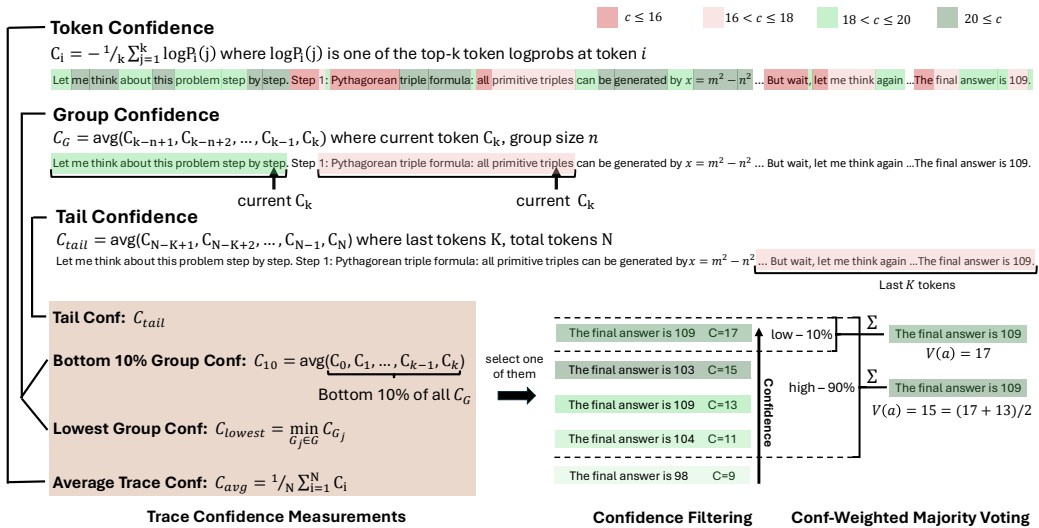

Figure 3: Confidence measurements and offline thinking with confidence.

where $I\{\cdot\}$ is the indicator function. The final answer is selected as the one with the highest vote count:

$$\hat{a} = \arg\max_a V(a).$$

**Confidence-Weighted Majority Voting.** Instead of treating each trace vote equally, we weight each final answer by the confidence of the associated trace. For every candidate answer $a$, we define its total vote weight as:

$$V(a) = \sum_{t \in T} C_t \cdot I(\text{answer}(t) = a),$$

where $C_t$ is the trace-level confidence chosen from the confidence measures discussed above. We select the answer with the highest weighted vote. This voting scheme favors answers supported by high-confidence traces, thereby reducing the impact of uncertain or low-quality reasoning answers.

**Confidence Filtering.** We apply confidence filtering in addition to weighted majority voting to control concentration on high-confidence reasoning traces. Confidence filtering selects the top-$\eta$ percent of traces based on trace confidence, ensuring only the most reliable paths contribute to the final answer. We provide two options across all confidence measurements: $\eta = 10\%$ and $\eta = 90\%$.

The top 10% option focuses on highest confidence scores, suitable when few reliable traces are expected to yield accurate results. However, relying on very few traces risks incorrect answers if the model exhibits bias. The top 90% option offers a more balanced approach, maintaining diversity and reducing model bias by including a broader range of traces. This ensures alternative reasoning paths are considered, especially when confidence distributions tend to be uniform. Fig. 3 provides illustration for confidence measurements and how offline thinking works with confidence. In addition, Alg. 1 provides the details of the algorithm.

### 3.3 ONLINE THINKING WITH CONFIDENCE

Evaluating confidence during online thinking enables real-time estimation of trace quality during generation, allowing dynamic termination of unpromising traces. This approach is particularly valuable in resource-constrained environments or when quick responses are necessary. The lowest group confidence metric can be effectively applied in this online setting. We can halt trace generation when token group confidence falls below a critical threshold, ensuring such traces would likely be excluded during confidence filtering.

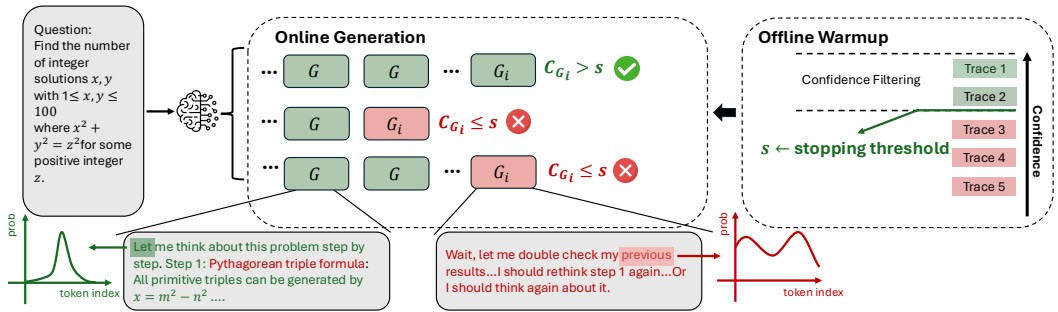

Figure 4: DeepConf during online generation.

We propose DeepConf-low and DeepConf-high, two algorithms based on lowest group confidence that adaptively stop generation and adjust trace budgets during online thinking. The approach includes two main components: offline warmup and adaptive sampling.

**Offline Warmup.** DeepConf requires an offline warmup phase to establish the stopping threshold $s$ for online determination. For each new prompt, we generate $N_{\text{init}}$ reasoning traces (e.g., $N_{\text{init}} = 16$). The stopping threshold $s$ is defined as: $s = \text{Percentile}_{100-\eta}(\{C_t : t \in T_{\text{warmup}}\})$, where $T_{\text{warmup}}$ represents all warmup traces, $C_t$ is the confidence of trace $t$, and $\eta$ is the desired keeping ratio. Specifically, DeepConf-low uses top $\eta = 10\%$ (corresponding to the 90th percentile) and DeepConf-high uses top $\eta = 90\%$ (corresponding to the 10th percentile) uniformly across all settings. This threshold ensures that during online generation, traces are terminated when their confidence falls below the level that retains the top $\eta\%$ highest-confidence traces from the warmup phase.

**Adaptive Sampling.** In DeepConf, we employ adaptive sampling across all methods to dynamically adjust the number of traces generated based on problem difficulty (Xue et al., 2023). Difficulty is assessed by consensus among generated traces, quantified by the ratio of majority vote weight $V(\hat{a})$ to total vote weight $\sum_a V(a)$: $\beta = \frac{V(\hat{a})}{\sum_a V(a)}$. $\tau$ is a preset consensus threshold. If $\beta < \tau$, the model does not reach consensus for the current problem, and trace generation continues until a fixed trace budget $B$ is met. Otherwise, trace generation halts, finalizing the answer with existing traces.

Since we use lowest group confidence, a sufficiently large warmup set yields an accurate estimate of the stopping threshold $s$; consequently, any trace terminated online has group confidence $< s$ and would be excluded by the offline filter. Thus the online procedure approximates the offline lowest-group-confidence policy, with accuracy approaching offline accuracy as $N_{init}$ increases (see Appendix C.2). We illustrate the online generation process in Fig. 4. In addition, Alg. 2 provides the details of the algorithm.

## 4 EXPERIMENTS

### 4.1 EXPERIMENTAL SETUP

**Models.** We evaluate five open-source reasoning LLMs from three model families: DeepSeek-8B[1] (Guo et al., 2025), Qwen3-8B, Qwen3-32B (Yang et al., 2025a), GPT-OSS-20B and GPT-OSS-120B (OpenAI, 2025). These models are recognized for strong mathematical reasoning and long-chain-of-thought performance, are fully open-source for reproducibility, and cover multiple parameter scales to test robustness. Complete generation hyperparameters and prompting templates are provided in Appendix G.

**Benchmarks.** We evaluate on five challenging datasets: AIME24 (Art of Problem Solving, 2024a;b), AIME25 (Art of Problem Solving, 2025a;b), BRUMO25 (bru, 2025), HMMT25 (HMMT,

---

[1]DeepSeek-8B refers to the Qwen3-8B model distilled from the DeepSeek-R1 (0528) model: https://huggingface.co/deepseek-ai/DeepSeek-R1-0528-Qwen3-8B.

---

**Algorithm 1:** Offline Thinking with Confidence

**Inputs:** Prompt $P$, number of traces $N$, filtering threshold $\eta$, confidence measurement $C(t)$
Initialize trace set $T \leftarrow \emptyset$, confidence set $C \leftarrow \emptyset$
**for** $i = 1$ to $N$ **do**
    Generate trace $t_i$ for prompt $P$, calculate trace confidence $C_i = C(t_i)$, and add $(t_i, C_i)$ to $(T, C)$
**end for**
Select top-$\eta$ percent of traces based on trace confidence $C_i$, compute $V(a)$ for all answers $a$
**return** Final answer $\hat{a}$ with highest weighted vote: $\hat{a} = \arg\max_a V(a)$

---

**Algorithm 2:** Online Thinking with Confidence (DeepConf-low/high)

**Inputs:** Prompt $P$, trace budget $B$, initial traces $N_{\text{init}}$, filtering threshold $\eta$, consensus threshold $\tau$
**Offline Warmup:**
Perform Algorithm 1 using $N = N_{\text{init}}$ traces with lowest group confidence
Compute threshold $s = \text{Percentile}_{100-\eta}(C_0, C_1, \cdots, C_{N_{\text{init}}-1})$ where we keep top-$\eta\%$ confident traces
Initialize trace set $T \leftarrow (t_0, t_1, \cdots, t_k)$, get vote values $V(a)$ for all answers $a$, and majority answer $\hat{a}$
**Online Generation:**
**while** $(V(\hat{a})/\sum_a V(a)) < \tau$ and $|T| < B$ **do**
    **while** generating trace $t$ **do**
        Generate next token $i$ and calculate group confidence $C_{G_i}$ for group $G_i$
        **If** $C_{G_i} < s$: stop generating trace $t$, **otherwise**: add token $i$ to trace $t$
    **end while**
    Add trace $t$ to $T$, compute trace confidence $C_t$, update vote counts $V(a)$, and majority answer $\hat{a}$
**end while**
**return** $\hat{a}$ and stop generation

---

2025), and GPQA (Rein et al., 2024). The first four are high-difficulty mathematical competition problems, while GPQA comprises graduate-level STEM reasoning tasks. All benchmarks are widely adopted in recent evaluations of top reasoning LLMs (e.g., Grok-4 (xAI, 2025), Qwen3 (Yang et al., 2025a), GPT-5 (OpenAI, 2025)) and featured in the MathArena leaderboard (Balunović et al., 2025).

**Baselines.** We adopt self-consistency (Wang et al., 2023) with majority voting as our primary baseline. Each LLM samples $T$ independent reasoning paths and selects the final answer via unweighted majority voting, as formalized in Sec. 3.2.

**Experimental Settings.** For each problem, we establish a common sampling frame by pre-generating a pool of 4,096 *complete* reasoning traces, serving as the foundation for offline and online evaluations. Offline experiments resample a working set of size $K$ (e.g., $K$=512) from it on each run and apply the specified voting method. Online experiments similarly resample a working set to drive *on-the-fly* generation with early stopping, ensuring consistent sampling across methods.

We report four key methods: (i) Pass@1 (single-trace accuracy), (ii) Cons@K (unweighted majority-vote accuracy with $K$ traces), (iii) Measure@K (confidence-weighted majority-vote accuracy), and (iv) Measure+top-$\eta\%$@K, which retains the top $\eta\%$ traces by confidence within the sampled working set before applying weighted majority voting (we use $\eta \in \{10, 90\}$). The specific confidence measure varies by setting. We also report total generated tokens. All metrics are averaged over 64 independent runs with fresh resampling; unless noted, tokens are counted end-to-end for all generated traces, with early-terminated traces contributing only tokens produced before stopping.

For online evaluation, we instantiate DeepConf-low and DeepConf-high using *Lowest Group Confidence* (Eq. 6) with an overlapping window of 2,048 tokens. Each problem begins with $N_{\text{init}}$=16 complete traces for offline warmup; we then set a run-specific stopping threshold $s = \min_{t \in T_{\text{top}}} C_t$, where $T_{\text{top}}$ contains the top-percentile traces by confidence ($\eta$=10 for *DeepConf-low*, $\eta$=90 for *DeepConf-high*; Sec. 3.3). During generation, traces whose current group confidence falls below $s$ are terminated early; completed traces are aggregated with confidence-weighted majority voting and generation stops adaptively once consensus $\geq \tau$ or budget $K$ is reached.

For offline evaluation, we benchmark three kinds of trace-level confidence from Sec. 3.1: (i) *Average Trace Confidence* (Eq. 3), (ii) *Bottom-10% Group Confidence* (Eq. 5), and (iii) *Tail Confidence* over the last 2,048 tokens (Eq. 7). For each metric we report Measure@K and Measure+top-$\eta\%$@K with $\eta \in \{10, 90\}$, where the top-$\eta\%$ cutoff is recomputed *within* the sampled set on every run (Sec. 3.2).

Table 1: Benchmarking confidence measurements in offline setting. Accuracy (%) is reported. Cons@512 and mean@512 denotes majority voting and average mean confidence using 512 traces. All experiments are repeated 64 times.

| Model | Dataset | Pass @1 | Cons @512 | Mean @512 | Bottom-10 Conf @512 | | Tail Conf @512 | |
|---|---|---|---|---|---|---|---|---|
| Retention Ratio | | | | | 90% | 10% | 90% | 10% |
| DeepSeek-8B | AIME24 | 83.0 | 86.7 | 86.7 | 86.7 | **93.3** | 86.7 | **93.3** |
| | AIME25 | 76.9 | 82.3 | 82.3 | 81.0 | **87.5** | 81.3 | 87.4 |
| | BRUMO25 | 80.0 | 93.2 | **93.3** | 93.3 | 93.3 | 93.3 | 93.3 |
| | HMMT25 | 58.1 | 69.6 | 69.9 | 69.9 | 79.5 | 69.9 | **83.9** |
| | GPQA-D | 62.8 | 72.5 | 72.5 | 71.2 | 70.6 | 72.8 | **74.0** |
| Qwen3-32B | AIME24 | 80.6 | 85.3 | 85.7 | 86.0 | **90.8** | 86.8 | 89.4 |
| | AIME25 | 71.7 | 80.1 | 80.0 | 80.1 | **80.2** | 80.1 | **80.2** |
| | BRUMO25 | 78.0 | **93.3** | **93.3** | 93.3 | 93.3 | 93.3 | 91.2 |
| | HMMT25 | 51.9 | 63.3 | 63.3 | 63.2 | 63.3 | **63.4** | 62.9 |
| | GPQA-D | 68.9 | 72.2 | 72.3 | 70.0 | 70.0 | **72.8** | 72.5 |
| GPT-OSS-120B | AIME24 | 91.9 | 96.7 | 96.7 | 96.3 | 96.5 | 96.7 | **97.4** |
| | AIME25 | 91.8 | 97.0 | 97.1 | 96.9 | 98.1 | 97.8 | **99.9** |
| | BRUMO25 | 75.6 | 86.7 | 86.8 | 85.3 | 82.9 | **89.9** | 89.4 |
| | HMMT25 | 78.9 | **92.9** | **92.9** | 92.9 | 90.5 | 92.9 | 88.9 |

## 4.2 OFFLINE EVALUATIONS

We present offline results with three models on five datasets at voting size $K{=}512$ in Table 1. We compare the following: Pass@1 = single-trace accuracy; Cons@512 = unweighted majority voting with 512 traces; Mean Conf@512 = confidence-weighted majority voting using average trace confidence (Eq. 3); Bottom-10% Conf@512 and Tail Conf@512 = confidence-weighted majority voting using (i) the mean of the lowest 10% group confidences (Eq. 5) and (ii) the mean confidence over the final 2k tokens (Eq. 7), respectively. The 90%/10% subcolumns indicate the retention ratio $\eta$ in confidence filtering: we retain the top $\eta\%$ highest-confidence traces within the sampled set before voting. For example, with $K{=}512$ and $\eta{=}10\%$, we keep approximately 51 traces for voting.

Overall, confidence-aware weighting with filtering consistently outperforms majority voting (Cons@512) across most settings. Filtering with $\eta{=}10\%$ yields the largest gains, with notable improvements like DeepSeek-8B on AIME25 (82.3% → 87.4%) and Qwen3-32B on AIME24 (85.3% → 90.8%); GPT-OSS-120B even reaches 99.9% on AIME25. Both local (Tail and Bottom-10%) and global (Average Trace) confidence measures show promising results in identifying confident traces. However, filtering involves important trade-offs: while aggressive filtering ($\eta{=}10\%$) maximizes accuracy gains in most cases, it can sometimes hurt performance due to model overconfidence on incorrect problems (e.g., GPT-OSS-120B). In such cases, conservative filtering ($\eta{=}90\%$) provides a safer option. Substantial improvements over pass@1 are observed across all methods, confirming the value of ensemble approaches. Detailed confidence measure comparisons are in Appendix C.4.

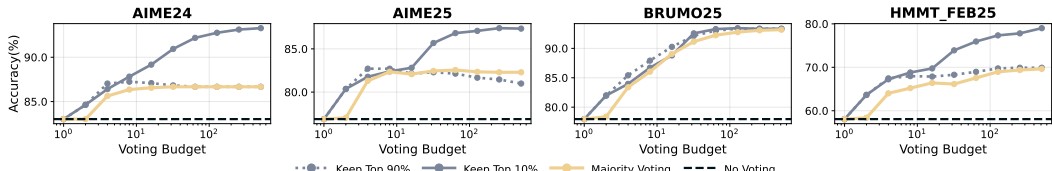

Figure 5: Offline accuracy with Lowest Group Confidence filtering (DeepSeek-8B) on AIME24, AIME25, BRUMO25, and HMMT25. The $\eta\%$ variant retains only the top $\eta\%$ highest-confidence traces before confidence-weighted majority voting.

We then show that Lowest Group Confidence is also effective. Fig. 5 reports offline results using Lowest Group Confidence (Eq. 6) to capture the least-confident token group (window size 2,048)

within each trace. Within each sampled working set we retain the top $\eta\%$ highest-confidence traces and then apply confidence-weighted majority voting. Across AIME24, AIME25, BRUMO25, and HMMT25 with DeepSeek-8B, retaining the top $\eta$=10% yields consistent gains over the best accuracy achieved by majority voting: +0.26 to +9.38 percentage points (average +5.27), and large improvements over single-trace (or no voting) accuracy ( +10.26 to +20.94 percentage points; average +14.30). The conservative $\eta$=90% setting matches or slightly exceeds the best majority-voting accuracy on all four datasets ( +0.16 to +0.57 percentage points; average +0.29) while still providing substantial improvements over single-trace accuracy (average +9.31). These results motivate the online variant: focusing on the least-confident segment reliably identifies traces with localized reasoning breakdowns, providing a strong signal for offline filtering and a natural stopping criterion during online generation. Beyond these results, we ablate the retention rate $\eta$ in Appendix C.3 and present the full offline results in Appendix D.

Table 2: Benchmark DeepConf in online setting. Accuracy (%) and tokens ($\times 10^8$) at voting size budget 512 for Majority Voting and DeepConf (high/low).

| Model | Dataset | Cons@512 | | DeepConf-high | | DeepConf-low | |
|---|---|---|---|---|---|---|---|
| | | Token | Acc | Token ($\Delta\%$) | Acc | Token ($\Delta\%$) | Acc |
| DeepSeek-8B | AIME24 | 3.55 | 86.7% | 1.45 (-59.0%) | 86.7% | 0.78 (-77.9%) | 92.5% |
| | AIME25 | 4.01 | 82.3% | 2.37 (-40.9%) | 81.4% | 1.24 (-69.0%) | 86.4% |
| | BRUMO25 | 3.56 | 93.3% | 2.17 (-39.2%) | 93.3% | 1.07 (-70.0%) | 93.3% |
| | HMMT25 | 4.49 | 69.8% | 3.43 (-23.5%) | 70.0% | 1.60 (-64.4%) | 77.6% |
| Qwen3-32B | AIME24 | 2.00 | 84.8% | 0.88 (-56.0%) | 86.4% | 0.66 (-66.8%) | 89.5% |
| | AIME25 | 2.43 | 80.1% | 1.61 (-33.7%) | 80.2% | 1.14 (-52.9%) | 80.2% |
| | BRUMO25 | 2.17 | 93.3% | 1.37 (-37.1%) | 93.3% | 0.96 (-55.7%) | 92.4% |
| | HMMT25 | 2.76 | 63.4% | 2.24 (-18.8%) | 63.6% | 1.55 (-43.8%) | 64.5% |
| GPT-OSS-120B | AIME24 | 2.66 | 96.7% | 1.20 (-54.6%) | 96.7% | 0.53 (-79.9%) | 97.0% |
| | AIME25 | 3.23 | 97.1% | 1.42 (-56.0%) | 97.0% | 0.49 (-84.7%) | 97.9% |
| | BRUMO25 | 2.68 | 83.8% | 1.81 (-32.6%) | 84.0% | 0.73 (-72.8%) | 83.4% |
| | HMMT25 | 4.09 | 92.8% | 2.78 (-32.0%) | 93.0% | 0.97 (-76.2%) | 92.0% |

## 4.3 ONLINE EVALUATIONS

We evaluate accuracy-cost trade-offs of the online algorithm by varying the budget $K \in \{32, 64, 128, 256, 512\}$, where *cost* counts all generated tokens, including partial tokens from early-stopped traces. Following Sec. 3.3, we perform a warm-up with $N_{\text{init}}$=16 traces to set the *stopping threshold s* using *Lowest Group Confidence* (LGC) of window size 2k: we set $s$ over the top-$\eta\%$ warm-up traces by confidence ($\eta \in \{10, 90\}$) and then terminate any new trace once its current group confidence drops below $s$. After each new trace completes, we reapply the same threshold $s$ for filtering so that the procedure matches the offline version of LGC filter while saving the cost of early-stopped traces. We consider two online variants: DeepConf-low ($\eta$=10%) and high ($\eta$=90%), which continue sampling until the consensus $\geq \tau$ (we use $\tau$=0.95) or the budget cap $K$ is reached. We compare with budget-only variants (always running to cap $K$) and more in Appendix C.1.

Table 2 shows the performance of the adaptive sampling version of DeepConf at the voting-size budget of $K$=512 on DeepSeek-8B, Qwen3-32B, and GPT-OSS-120B. Compared with the majority voting baseline, DeepConf-low reduces tokens by 43-79% across AIME24, AIME25, BRUMO25, and HMMT25. While it matches or improves accuracy in most cases (e.g., DeepSeek-8B AIME24: +5.8%), it experiences accuracy drops in a few settings (e.g., Qwen3-32B BRUMO25: $-0.9\%$). The more conservative DeepConf-high saves 18-59% tokens on these sets while maintaining accuracy or incurring only minimal performance degradation. Fig. 6 visualizes the token reduction patterns for GPT-OSS-120B, showing how DeepConf achieves substantial computational savings (i.e., up to 85.8%) while preserving competitive accuracy across different math reasoning tasks.

Fig. 7 compares DeepConf with majority voting on DeepSeek-8B. DeepConf methods show clear efficiency advantages while maintaining equal accuracy: DeepConf-low achieves token savings of 62.88% and DeepConf-high 47.67% compared to the majority voting at the same accuracy levels. In terms of performance, DeepConf's behavior mirrors the offline setting: $\eta$=10% (low) filtering yields

the highest accuracy gains in most cases, though it may occasionally result in accuracy drops on specific datasets (e.g., GPT-OSS-120B on HMMT25 in Table 2). These results support our design: using the least-confident segment to gate traces provides a strong, local signal for early stop, and the adaptive consensus stop further saves tokens without sacrificing accuracy. Besides, we provide an ablation of the warm-up size $N_{init}$ in Appendix C.2 and the full online results in Appendix E.

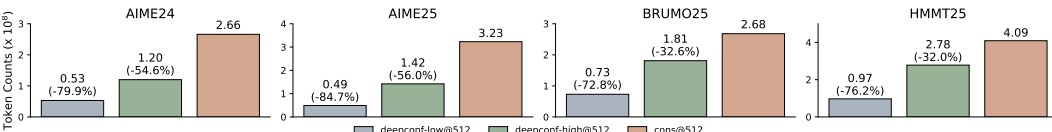

Figure 6: Generated tokens comparison across different tasks based on GPT-OSS-120B

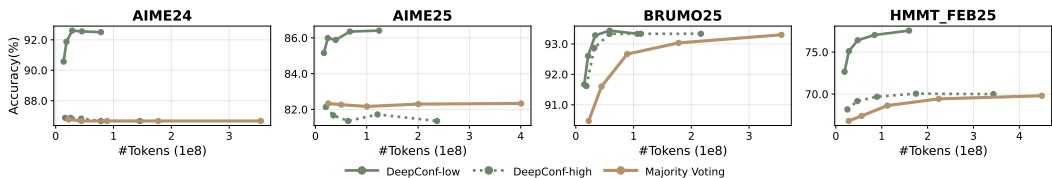

Figure 7: Accuracy vs. generated tokens for online Lowest Group Confidence filtering (DeepSeek-8B) on AIME24, AIME25, BRUMO25, and HMMT25. High/low means keeping the traces with top 90%/10% confidence for voting.

## 5    RELATED WORK

This section focuses on confidence estimation, which is most central to our approach. For a broader review of literature on test-time scaling and efficient reasoning, please refer to Appendix A. Confidence estimation techniques offer a complementary direction by directly quantifying the reliability of model outputs. A growing body of work proposes metrics such as token-level entropy and uncertainty scores (Fadeeva et al., 2024), self-certainty based on KL divergence from a uniform distribution (Kang et al., 2025), and specialized confidence tokens learned during fine-tuning (Chuang et al., 2024). In the same spirit, Dynasor (Fu et al., 2024) uses semantic entropy (Farquhar et al., 2024) as a confidence signal. These signals have been applied to re-ranking (Jain et al., 2024), selective generation (Ren et al., 2023), and abstention in high-stakes domains (Han et al., 2024), and they consistently exhibit better calibration than raw sequence probabilities (Geng et al., 2024). Wang & Zhou (2024) applies confidence at the answer position to elicit model reasoning ability without prompting. Recent work further employs various confidence metrics as reward signals for reinforcement learning (Zhao et al., 2025; Zuo et al., 2025; Zhang et al., 2025; Prabhudesai et al., 2025; Gao et al., 2025; Agarwal et al., 2025), showing improved model performance. Yang & Holtzman (2025) note that entropy drops late in generation, and earlier branching yields higher quality samples.

Integrating confidence into test-time reasoning provides a way to assess the quality of individual traces before aggregation. Recent results with *global* confidence, which is computed at the sequence level and applied post hoc to rank or select among completed candidates, show that combining multi-sample reasoning with confidence-aware selection can outperform majority voting while using fewer generated tokens (Kang et al., 2025). In contrast, DeepConf relies on a lightweight *local* confidence signal that is updated along each trajectory and triggers only-the-fly pruning of low-confidence traces, yielding more token-efficient parallel generation and higher accuracy.

## 6    CONCLUSION

We present DeepConf, a simple yet effective method that significantly enhances both reasoning performance and computational efficiency in ensemble voting scenarios. Through extensive experiments across state-of-the-art reasoning models and challenging datasets, DeepConf demonstrates substantial accuracy improvements while achieving meaningful token savings, with consistent benefits observed across model scales from 8B to 120B parameters. We hope this method shows the potential of a practical and scalable solution for efficient LLM reasoning.

## 7 ACKNOWLEDGMENTS

This work was primarily conducted during Yichao Fu's internship at Meta FAIR. We gratefully acknowledge the Meta FAIR team for providing the essential computational resources and infrastructure that made this research possible. We also thank the open-source community for developing and maintaining the foundational models and serving frameworks utilized in our experiments.

## 8 REPRODUCIBILITY STATEMENT

We provide full implementation details to facilitate reproduction. Section 3 presents complete algorithmic descriptions with pseudocode (Algorithm 1-2). All hyperparameters are documented in Tables 11-12 (Appendix G), including model-specific generation settings, warmup configurations, and filtering thresholds. Source code with detailed vLLM integration instructions is available at https://github.com/facebookresearch/deepconf and as supplementary materials. Experimental protocols, including 64-run repetitions and evaluation procedures, are detailed in Section 4.1. Dataset sources and prompt templates are specified in Section 4.1 and Appendix G.

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

# A EXTENDED RELATED WORK

## A.1 TEST TIME SCALING

Current LLMs increasingly succeed by allocating very large amounts of reasoning at inference, a paradigm we call test-time scaling (Snell et al., 2024; Welleck et al., 2024). Along one axis, Chain-of-Thought (Wei et al., 2022) depth is scaled by lengthening a single reasoning trajectory through more thinking steps; representative models include o1 (Jaech et al., 2024), DeepSeek R1 (Guo et al., 2025), Kimi K1.5 (Team et al., 2025), Qwen3 (Yang et al., 2025a), and Grok-4 (xAI, 2025), which typically rely on large-scale RL with lots of samples, as well as simpler fine-tuning approaches such as STILL-2 (Min et al., 2024), s1 (Muennighoff et al., 2025), and LIMO (Ye et al., 2025). Along a complementary axis, parallel generation is scaled by increasing the number of trajectories and aggregating them: Self-Consistency (Wang et al., 2023) and Best-of-N (Brown et al., 2024; Irvine et al., 2023) sample multiple candidates and select via voting or a score, while REBASE (Wu et al., 2024) expands breadth with tree-structured exploration. To optimize these allocations, DVTS (Beeching et al.) and DORA (Wang et al., 2025c) introduce adaptive PRM-guided search and direction-oriented strategies to maximize accuracy under fixed compute budgets. Chen et al. (2024a) analyze parallel generation in compound AI systems. These two axes can be combined to trade compute for accuracy under deployment constraints, and they underpin many recent reasoning-centric systems.

## A.2 EFFICIENT REASONING

Test-time scaling for reasoning seeks better accuracy-compute trade-offs through adaptive sampling and richer aggregation. On the parallel axis, Early-Stopping Self-Consistency (ESC), Difficulty-Adaptive Self-Consistency (DSC), Reasoning-Aware Self-Consistency (RASC), Adaptive-Consistency, Dynamic Voting, and Dynasor achieve more efficient self-consistency by reducing the required sample count while preserving accuracy (Li et al., 2024; Wan et al., 2025; Aggarwal et al., 2023; Xue et al., 2023; Fu et al., 2024; Wang et al., 2025b). On the CoT-depth axis, efficient CoT fine-tuning methods elicit shorter, more efficient chains (Chen et al., 2024b; Luo et al., 2025; Hou et al., 2025), whereas Dynasor-CoT (Fu et al., 2025) and DEER (Yang et al., 2025b) optimize inference without additional training. Other works refine aggregation: ranked voting (Wang et al., 2025a) collects ranked candidate lists for more nuanced preference aggregation, likelihood-weighted scoring (Soft-SC) (Wang et al., 2024) leverages model probabilities when no single answer dominates, and verification-augmented voting filters logically inconsistent paths using external tools (Toh et al., 2024). Complementarily, Hassid et al. (2025) show that preferring shorter CoT chains among multiple samples can improve accuracy. DeepConf leverages local confidence to improve accuracy by filtering out low-confidence traces; in online generation, it further performs early termination when local confidence drops below threshold, reducing token usage.

# B FUTURE WORK

We believe several promising directions emerge from this work. First, extending DeepConf to reinforcement learning settings could leverage confidence-based early stopping to guide policy exploration and improve sample efficiency during training. Second, addressing cases where models exhibit high confidence on incorrect reasoning paths , which is a key limitation observed in our experiments. Future work can also explore more robust confidence calibration techniques and uncertainty quantification methods to better identify and mitigate overconfident but erroneous predictions.

# C ABLATION STUDY

## C.1 ABLATION ON CONSENSUS THRESHOLDS

We ablate the consensus threshold $\tau$ in online Algorithm 2, using budget-only (not using $(V(\hat{a})/\sum_a V(a)) < \tau$ to do early stopping) as baseline in Table 3. After generating $N_{\text{init}}$=16 warmup traces, we check modal agreement before each new trace generation and stop generating more samples for this problem if the agreement exceeds $\tau$. We evaluate on Qwen3-32B with AIME24. $\tau$=0.95 achieves optimal balance: it preserves accuracy exactly while saving

15.4%/52.8% tokens at $B = 32/512$ (DeepConf-low) and 22.0%/54.7% (DeepConf-high). Conservative $\tau=1.0$ weakens savings without accuracy drop. More aggressive thresholds ($\tau=0.90, 0.85$) increase savings up to 69.6% but cause accuracy drops in DeepConf-low (e.g., $-0.26$pp at $\tau=0.90$), while DeepConf-high maintains perfect accuracy even at $\tau=0.85$, showing greater robustness. Because DeepConf-high already retains a larger pool of traces by design, changing $\tau$ has a smaller influence on the final vote. Besides, Token savings are larger at $B = 512$ than at $B = 32$ because adaptive stopping only applies after the $N_{init} = 16$ warm-up traces. At $B = 512$ it can truncate up to the remaining 496 generations, whereas at $B = 32$ it can eliminate at most 16. As a result, the same rate of early terminations translates into far greater relative token reductions at higher budgets. Overall, $\tau=0.95$ offers the best tradeoff, cutting tokens by over half with zero accuracy loss.

Table 3: Ablation on adaptive thresholds (Qwen3-32B @ AIME24). We report accuracy and token usage at voting budgets $B \in \{32, 512\}$. Accuracy deltas are in percentage points (p.p.) and token deltas are percent changes, both measured relative to the budget-only baseline (no adaptive early stopping) at the same $B$. Token counts are shown in scientific notation.

|  |  | $B = 32$ | | $B = 512$ | |
|---|---|---|---|---|---|
|  |  | Accuracy ($\Delta$ p.p.) | Tokens ($\Delta$ %) | Accuracy ($\Delta$ p.p.) | Tokens ($\Delta$ %) |
| **High** | *Baseline* | *87.50%* | *1.23e7* | *86.35%* | *1.94e8* |
|  | $\tau = 1.0$ | 87.50% (+0.00) | 1.03e7 (-16.5%) | 86.35% (+0.00) | 1.29e8 (-33.4%) |
|  | $\tau = 0.95$ | 87.50% (+0.00) | 9.59e6 (-22.0%) | 86.35% (+0.00) | 8.79e7 (-54.7%) |
|  | $\tau = 0.90$ | 87.50% (+0.00) | 8.88e6 (-27.7%) | 86.35% (+0.00) | 7.13e7 (-63.2%) |
|  | $\tau = 0.85$ | 87.50% (+0.00) | 8.39e6 (-31.8%) | 86.35% (+0.00) | 6.01e7 (-69.0%) |
| **Low** | *Baseline* | *87.92%* | *1.05e7* | *89.48%* | *1.40e8* |
|  | $\tau = 1.0$ | 87.92% (+0.00) | 8.97e6 (-15.0%) | 89.48% (+0.00) | 8.94e7 (-36.3%) |
|  | $\tau = 0.95$ | 87.92% (+0.00) | 8.92e6 (-15.4%) | 89.48% (+0.00) | 6.62e7 (-52.8%) |
|  | $\tau = 0.90$ | 87.66% (-0.26) | 8.13e6 (-22.9%) | 89.43% (-0.05) | 5.14e7 (-63.4%) |
|  | $\tau = 0.85$ | 87.45% (-0.47) | 7.75e6 (-26.5%) | 89.17% (-0.31) | 4.26e7 (-69.6%) |

## C.2 Ablation on Warmup Sampling Size.

Table 4 compares warmup sizes $N_{\text{init}} \in \{8, 16, 32\}$ with the budget-only online DeepConf method at voting budget $B = 512$ under the DeepConf-low setting (top $\eta = 10\%$ by confidence). Across models and datasets we observe that: Increasing $N_{\text{init}}$ stabilizes the empirical confidence distribution used to set the threshold $s$ and, in practice, generally makes online accuracy closer to the offline baseline (smaller $|\Delta\text{Acc}|$); however, the threshold-accuracy relationship is model- and dataset-dependent and need not improve monotonically. On tokens, the net effect is driven by two forces: (i) a larger fixed warm-up cost because all $N_{\text{init}}$ traces run to completion and (ii) the post-warm-up early-termination rate over the remaining $B - N_{\text{init}}$ traces. As these forces trade off, total token usage is also not necessarily monotone in $N_{\text{init}}$. Empirically, $|\Delta\text{Acc}|$ across warm-up sizes is small (typically $\leq 1.0$ p.p.). We therefore adopt $N_{\text{init}}=16$ as a balanced default: sufficiently close to the offline baseline while avoiding excessive warm-up overhead.

## C.3 Ablation on Filtering Percent.

Table 5 investigates the effect of varying the keeping percentage for filtering method using Lowest Group Confidence metric (group size 2,048). The retention ratio $\eta$ sweeping from top 90% to top 10%. For each model-dataset pair, we sweep voting sizes $B \in \{1, \ldots, 512\}$ in the offline setting and report the best accuracy attained. Across DeepSeek-8B, Qwen3-8B, and Qwen3-32B, more aggressive filtering (smaller top percentages, retaining fewer traces) generally yields higher accuracy in most cases, though the optimal $\eta$ can vary by dataset. For instance, top 10% frequently achieves the best performance, but top 25% or top 50% may be optimal for certain model-dataset combinations, indicating that the ideal filtering threshold depends on task characteristics. Mechanistically, the filter preferentially discards low-confidence (and often incorrect) traces, concentrating the vote on higher-confidence evidence and thereby improving accuracy on average.

Table 4: Impact of warmup size $N_{init}$ at fixed voting budget $B = 512$ (online, DEEPCONF-Low). Each online cell reports Accuracy (%) with $\Delta$ in p.p. and Token usage ($\times 10^8$) with relative $\Delta$ in %, both described as relative to the *offline* baseline at $B = 512$ (keep top $\eta = 10\%$; lowest group confidence). Boldface marks the warm-up size whose online accuracy is closest to the offline baseline (smallest $|\Delta \text{Acc}|$). The last column, labeled *Offline*, reports the baseline (Accuracy / Token) at $B = 512$.

| Model | Dataset | $N_{init} = 8$ | $N_{init} = 16$ | $N_{init} = 32$ | Offline |
|---|---|---|---|---|---|
| DeepSeek-8B | AIME24 | 92.2% (-1.0) / 1.60 (-54.9%) | 92.5% (-0.8) / 1.51 (-57.4%) | **92.9% (-0.4) / 1.52 (-57.1%)** | 93.3% / 3.55 |
| | AIME25 | 86.0% (-1.4) / 1.94 (-51.7%) | 86.4% (-0.9) / 1.85 (-53.8%) | **86.7% (-0.7) / 1.85 (-53.8%)** | 87.3% / 4.01 |
| | BRUMO25 | **93.3% (+0.0) / 1.63 (-54.1%)** | 93.3% (+0.0) / 1.55 (-56.5%) | 93.4% (+0.1) / 1.54 (-56.6%) | 93.3% / 3.56 |
| | GPQA | 71.8% (-0.1) / 4.93 (-50.3%) | 71.7% (-0.2) / 4.75 (-52.1%) | **71.9% (-0.0) / 4.78 (-51.8%)** | 71.9% / 9.92 |
| | HMMT25 | 76.8% (-2.2) / 2.05 (-54.4%) | 77.6% (-1.5) / 1.91 (-57.4%) | **78.2% (-0.8) / 1.89 (-57.8%)** | 79.0% / 4.49 |
| Qwen3-32B | AIME24 | 89.2% (-0.9) / 1.43 (-28.2%) | 89.5% (-0.6) / 1.40 (-29.7%) | **90.1% (-0.1) / 1.39 (-30.2%)** | 90.1% / 2.00 |
| | AIME25 | 80.5% (+0.3) / 1.72 (-29.2%) | **80.2% (+0.0) / 1.69 (-30.6%)** | 80.1% (-0.1) / 1.68 (-30.7%) | 80.2% / 2.43 |
| | BRUMO25 | **92.8% (+0.1) / 1.53 (-29.5%)** | 92.4% (-0.3) / 1.49 (-31.4%) | 92.4% (-0.3) / 1.49 (-31.4%) | 92.8% / 2.17 |
| | GPQA | 72.9% (+0.3) / 5.87 (-21.1%) | 73.0% (+0.3) / 5.77 (-22.6%) | **72.9% (+0.2) / 5.74 (-22.9%)** | 72.7% / 7.44 |
| | HMMT25 | 65.4% (+1.0) / 1.94 (-29.8%) | **64.5% (+0.1) / 1.90 (-31.3%)** | 64.6% (+0.2) / 1.89 (-31.5%) | 64.4% / 2.76 |
| Qwen3-8B | AIME24 | 86.5% (-0.2) / 1.53 (-34.1%) | 86.5% (-0.2) / 1.50 (-35.3%) | **86.7% (+0.1) / 1.51 (-35.0%)** | 86.7% / 2.32 |
| | AIME25 | 78.5% (+0.6) / 1.83 (-33.9%) | **78.1% (+0.2) / 1.78 (-35.6%)** | 78.2% (+0.3) / 1.79 (-35.3%) | 77.9% / 2.77 |
| | BRUMO25 | **82.6% (+0.2) / 1.67 (-34.4%)** | 82.7% (+0.3) / 1.63 (-35.9%) | 82.9% (+0.5) / 1.63 (-36.1%) | 82.4% / 2.54 |
| | GPQA | 65.1% (-0.4) / 4.92 (-34.1%) | 65.2% (-0.3) / 4.82 (-35.5%) | **65.3% (-0.2) / 4.81 (-35.5%)** | 65.5% / 7.47 |
| | HMMT25 | 62.3% (-0.8) / 1.98 (-36.2%) | 62.7% (-0.4) / 1.94 (-37.5%) | **63.0% (-0.2) / 1.94 (-37.5%)** | 63.1% / 3.10 |

Table 5: Best accuracy (%) across different filter sizes using Lowest Group Confidence with group size 2048

| Model | Dataset | Maj. | Top90 | Top75 | Top50 | Top25 | Top10 |
|---|---|---|---|---|---|---|---|
| DeepSeek-8B | AIME24 | 86.7% | 87.2% | 87.7% | 88.9% | 91.8% | **93.3%** |
| | AIME25 | 82.6% | 82.7% | 82.8% | 83.6% | 85.9% | **87.4%** |
| | BRUMO25 | 93.2% | 93.3% | 93.3% | 93.3% | 93.3% | **93.4%** |
| | HMMT25 | 69.6% | 69.9% | 70.3% | 73.4% | 75.4% | **79.0%** |
| Qwen3-8B | AIME24 | 81.4% | 82.1% | 82.2% | 84.9% | **86.9%** | 86.7% |
| | AIME25 | 82.6% | **82.7%** | 81.8% | 79.5% | 79.4% | 77.9% |
| | BRUMO25 | 81.5% | 81.8% | 82.3% | **83.3%** | **83.3%** | 82.4% |
| | HMMT25 | 60.1% | 60.3% | 60.2% | 60.6% | 61.6% | **63.1%** |
| Qwen3-32B | AIME24 | 87.8% | 88.3% | 88.9% | 88.9% | 89.1% | **90.2%** |
| | AIME25 | 80.5% | 80.4% | 80.5% | 81.0% | **81.5%** | 80.9% |
| | BRUMO25 | **93.3%** | **93.3%** | **93.3%** | **93.3%** | **93.3%** | 92.8% |
| | HMMT25 | 63.4% | 64.0% | 64.5% | 66.7% | **67.3%** | 64.4% |

## C.4 ABLATION ON CONFIDENCE METRICS

In this offline ablation, we report the accuracy at voting size $B = 512$ for each model-dataset pair (Tables 6, 7, 8 and 9). We compare aggregation rules that differ only in how they compute a per-trace confidence score and whether they do filtering before voting. Maj. represents standard majority voting without confidence weighting. Mean computes the average trace confidence over the entire trace and uses it to weight votes. L(w) denotes Lowest Group Confidence with group size w. B(q%) represents Bottom Percent Confidence, which retains the bottom q% least confident groups and computes the average confidence from these selected groups for voting weights. For positional confidence computation, Head(q%) calculates confidence using only the first q% of tokens in each trace, while Tail(q%) computes confidence from the last q% of tokens. Tail(2k) uses a fixed window of the last 2,048 tokens regardless of the total trace length. The @$\eta$% suffix indicates a filtering mechanism that retains only the top $\eta$% of traces ranked by confidence before applying the respective voting method. For example, Mean@10% first selects the top 10% most confident traces and then applies confidence-weighted voting, while Tail(2k)@90% keeps the top 90% of traces based on tail confidence before voting.

Across models and datasets, head-based confidence has weak correlation with final correctness and typically matches plain majority voting; applying head-based filters even hurts on average, reflecting that early tokens are dominated by setup, paraphrase, and exploratory planning with little discriminative signal. By contrast, tail- and mean-based signals frequently yield gains. Notably, on DeepSeek-8B AIME25, a tail variant reaches 89.6% with only an 8B model, and on GPT-OSS-120B AIME25, Tail(2k)@10% attains 99.9%. Conceptually, the Lowest Group Confidence (LGC) metric is an ex-

treme case of Bottom-Percent confidence (it takes the minimum over sliding groups), yet it remains competitive: over 23 model-dataset pairs, LGC (2k window) with a 10% filter averages 84.4%, compared to 84.0% for Bottom-10%@10% and 84.0% for Bottom-50%@10%, and higher than Mean10% at 83.9%. Tail-focused variants are especially reliable defaults, with Tail(10%)@10% and Tail(2k)@10% averaging 84.6% and 84.5%, respectively. Overall, local confidence signals, including tail, bottom, and lowest, are not inferior to global average trace confidence and, on average, deliver equal or better accuracy across settings.

Table 6: Accuracy (%) at voting size = 512. Maj.=majority; Mean: average trace confidence; @$\eta$=keep top $\eta$% by confidence. Head (10%): first 10% tokens.

| Model | Dataset | Maj. | Mean | Mean@10 | Mean@90 | Head (10%) | Head (10%)@10 | Head (10%)@90 |
|---|---|---|---|---|---|---|---|---|
| DeepSeek-8B | AIME24 | 86.7% | 86.7% | 93.3% | 86.7% | 86.7% | 86.7% | 86.7% |
| | AIME25 | 82.3% | 82.3% | 88.6% | 80.7% | 82.2% | 81.1% | 80.9% |
| | BRUMO25 | 93.2% | 93.3% | 93.4% | 93.3% | 93.2% | 91.2% | 93.2% |
| | HMMT25 | 69.6% | 69.9% | 84.3% | 69.9% | 69.6% | 69.1% | 69.7% |
| | GPQA | 72.5% | 72.5% | 71.6% | 72.7% | 72.5% | 70.7% | 72.5% |
| Qwen3-8B | AIME24 | 80.1% | 80.1% | 86.7% | 80.5% | 80.1% | 80.5% | 80.0% |
| | AIME25 | 82.6% | 82.7% | 74.0% | 82.1% | 82.7% | 73.8% | 82.4% |
| | BRUMO25 | 80.9% | 81.0% | 82.9% | 81.9% | 80.9% | 82.3% | 80.9% |
| | HMMT25 | 60.0% | 60.0% | 62.2% | 60.0% | 60.0% | 59.2% | 60.0% |
| | GPQA | 63.8% | 63.8% | 63.9% | 64.3% | 63.8% | 63.6% | 64.1% |
| Qwen3-32B | AIME24 | 85.3% | 85.7% | 93.2% | 86.5% | 85.3% | 84.1% | 86.1% |
| | AIME25 | 80.1% | 80.0% | 82.0% | 80.0% | 80.1% | 77.4% | 80.2% |
| | BRUMO25 | 93.3% | 93.3% | 93.3% | 93.3% | 93.3% | 92.6% | 93.3% |
| | HMMT25 | 63.3% | 63.3% | 62.4% | 63.3% | 63.3% | 59.0% | 63.2% |
| | GPQA | 72.2% | 72.3% | 73.3% | 72.6% | 72.2% | 72.2% | 72.2% |
| GPT-OSS-20B | AIME24 | 96.7% | 96.7% | 96.7% | 96.7% | 96.7% | 96.7% | 96.7% |
| | AIME25 | 95.3% | 95.3% | 94.0% | 94.8% | 95.4% | 94.9% | 94.8% |
| | BRUMO25 | 87.3% | 87.4% | 87.7% | 87.5% | 87.3% | 88.2% | 87.4% |
| | HMMT25 | 89.9% | 89.7% | 86.5% | 89.2% | 89.9% | 89.0% | 89.7% |
| GPT-OSS-120B | AIME24 | 96.7% | 96.7% | 96.7% | 96.7% | 96.7% | 97.7% | 96.7% |
| | AIME25 | 97.0% | 97.1% | 97.9% | 97.9% | 97.1% | 97.6% | 97.1% |
| | BRUMO25 | 86.7% | 86.8% | 85.7% | 87.6% | 86.7% | 85.9% | 86.7% |
| | HMMT25 | 92.9% | 92.9% | 80.3% | 93.0% | 92.9% | 90.6% | 93.0% |
| Average Acc. | | 83.0% | 83.0% | 83.9% | 83.1% | 83.0% | 81.9% | 82.9% |

Table 7: Accuracy (%) at voting size = 512. @$\eta$=keep top $\eta$% by confidence. Tail (2k): last 2,048 tokens; Tail (10%): last 10% tokens.

| Model | Dataset | Tail (10%) | Tail (10%)@10 | Tail (10%)@90 | Tail (2k) | Tail (2k)@10 | Tail (2k)@90 |
|---|---|---|---|---|---|---|---|
| DeepSeek-8B | AIME24 | 86.7% | 93.3% | 86.7% | 86.7% | 93.3% | 86.7% |
| | AIME25 | 82.6% | 89.6% | 81.6% | 82.4% | 87.4% | 81.3% |
| | BRUMO25 | 93.3% | 93.3% | 93.3% | 93.3% | 93.3% | 93.3% |
| | HMMT25 | 69.9% | 84.0% | 69.9% | 69.9% | 83.9% | 69.9% |
| | GPQA | 72.8% | 72.7% | 72.8% | 72.8% | 74.0% | 72.8% |
| Qwen3-8B | AIME24 | 80.3% | 87.1% | 80.7% | 80.4% | 86.7% | 80.7% |
| | AIME25 | 82.8% | 75.6% | 82.9% | 82.7% | 75.7% | 82.7% |
| | BRUMO25 | 80.9% | 81.6% | 81.6% | 80.9% | 81.4% | 81.5% |
| | HMMT25 | 60.0% | 64.1% | 60.0% | 60.0% | 63.8% | 60.0% |
| | GPQA | 63.9% | 64.1% | 64.2% | 63.8% | 65.7% | 64.4% |
| Qwen3-32B | AIME24 | 86.0% | 92.1% | 87.1% | 85.9% | 89.4% | 86.8% |
| | AIME25 | 80.0% | 86.6% | 80.1% | 80.0% | 80.2% | 80.1% |
| | BRUMO25 | 93.3% | 92.1% | 93.3% | 93.3% | 91.2% | 93.3% |
| | HMMT25 | 63.4% | 62.7% | 63.4% | 63.4% | 62.9% | 63.4% |
| | GPQA | 72.3% | 73.2% | 72.7% | 72.4% | 72.5% | 72.8% |
| GPT-OSS-20B | AIME24 | 96.7% | 96.7% | 96.7% | 96.7% | 96.7% | 96.7% |
| | AIME25 | 95.5% | 94.7% | 95.7% | 95.7% | 95.9% | 96.1% |
| | BRUMO25 | 87.3% | 88.0% | 87.1% | 87.3% | 84.6% | 87.1% |
| | HMMT25 | 89.7% | 85.3% | 89.9% | 90.1% | 88.2% | 89.8% |
| GPT-OSS-120B | AIME24 | 96.7% | 97.4% | 96.8% | 96.7% | 97.4% | 96.7% |
| | AIME25 | 97.3% | 99.4% | 97.6% | 97.4% | 99.9% | 97.8% |
| | BRUMO25 | 87.9% | 85.6% | 89.9% | 88.2% | 89.4% | 89.9% |
| | HMMT25 | 92.9% | 87.4% | 93.1% | 92.9% | 88.9% | 92.9% |
| Average Acc. | | 83.1% | 84.6% | 83.4% | 83.2% | 84.5% | 83.3% |

## D SCALING BEHAVIOR FOR OFFLINE DEEPCONF

The offline method applies confidence filtering followed by confidence-weighted majority voting (Sec. 3.2) under two retention settings using Lowest Group Confidence. We evaluate across 5 model

Table 8: Accuracy (%) at voting size = 512. Maj.=majority; L(x)=Lowest Group Confidence with a sliding window of x tokens; @$\eta$ keeps the top $\eta$% by the Lowest-confidence.

| Model | Dataset | Maj. | L(512) | L(1K) | L(2K) | L(512)@10 | L(1K)@10 | L(2K)@10 | L(2K)@90 |
|---|---|---|---|---|---|---|---|---|---|
| DeepSeek-8B | AIME24 | 86.7% | 86.7% | 86.7% | 86.7% | 92.8% | 93.1% | 93.3% | 86.7% |
| | AIME25 | 82.3% | 82.2% | 82.3% | 82.2% | 86.9% | 87.1% | 87.3% | 81.0% |
| | BRUMO25 | 93.2% | 93.3% | 93.3% | 93.3% | 93.3% | 93.3% | 93.3% | 93.3% |
| | HMMT25 | 69.6% | 69.9% | 69.9% | 69.9% | 77.3% | 77.8% | 79.0% | 69.9% |
| | GPQA | 72.5% | 72.5% | 72.5% | 72.5% | 71.7% | 71.7% | 71.9% | 72.5% |
| Qwen3-8B | AIME24 | 80.1% | 80.1% | 80.1% | 80.1% | 86.7% | 86.7% | 86.7% | 80.3% |
| | AIME25 | 82.6% | 82.7% | 82.7% | 82.6% | 74.2% | 76.6% | 77.9% | 82.7% |
| | BRUMO25 | 80.9% | 80.9% | 80.9% | 80.9% | 81.9% | 82.3% | 82.4% | 81.5% |
| | HMMT25 | 60.0% | 60.0% | 60.0% | 60.0% | 62.8% | 63.1% | 63.1% | 60.0% |
| | GPQA | 63.8% | 63.6% | 63.6% | 63.7% | 64.8% | 65.1% | 65.5% | 64.1% |
| Qwen3-32B | AIME24 | 85.3% | 85.3% | 85.2% | 85.6% | 87.3% | 86.8% | 90.1% | 86.2% |
| | AIME25 | 80.1% | 80.0% | 80.0% | 80.0% | 78.5% | 82.6% | 80.2% | 80.1% |
| | BRUMO25 | 93.3% | 93.3% | 93.3% | 93.3% | 87.2% | 88.8% | 92.8% | 93.3% |
| | HMMT25 | 63.3% | 63.3% | 63.4% | 63.4% | 62.6% | 64.7% | 64.4% | 63.4% |
| | GPQA | 72.2% | 72.3% | 72.4% | 72.4% | 73.5% | 72.8% | 72.7% | 72.8% |
| GPT-OSS-20B | AIME24 | 96.7% | 96.7% | 96.7% | 96.7% | 93.3% | 93.3% | 95.3% | 96.7% |
| | AIME25 | 95.3% | 95.4% | 95.4% | 95.4% | 96.2% | 96.2% | 96.0% | 95.5% |
| | BRUMO25 | 87.3% | 87.3% | 87.3% | 87.3% | 87.6% | 87.4% | 87.6% | 87.4% |
| | HMMT25 | 89.9% | 89.8% | 89.8% | 89.9% | 90.1% | 90.2% | 89.5% | 89.9% |
| GPT-OSS-120B | AIME24 | 96.7% | 96.7% | 96.7% | 96.7% | 96.6% | 97.2% | 97.0% | 96.7% |
| | AIME25 | 97.0% | 97.0% | 97.0% | 97.1% | 97.4% | 97.8% | 98.0% | 97.0% |
| | BRUMO25 | 86.7% | 86.7% | 86.8% | 86.8% | 85.5% | 86.3% | 85.8% | 86.8% |
| | HMMT25 | 92.9% | 92.9% | 93.0% | 92.9% | 92.6% | 92.0% | 92.0% | 93.1% |
| Average Acc. | | 83.0% | 83.0% | 83.0% | 83.0% | 83.5% | 84.0% | 84.4% | 83.1% |

Table 9: Accuracy (%) at voting size = 512. Maj.=majority; B(q%)=Bottom-q-Percent confidence within a 2,048-token sliding window; @$\eta$ keeps the top $\eta$% by the Bottom-confidence.

| Model | Dataset | Maj. | B(10%)@10 | B(10%)@90 | B(10%) | B(50%)@10 | B(50%)@90 | B(50%) |
|---|---|---|---|---|---|---|---|---|
| DeepSeek-8B | AIME24 | 86.7% | 93.3% | 86.7% | 86.7% | 93.3% | 86.7% | 86.7% |
| | AIME25 | 82.3% | 87.5% | 81.0% | 82.2% | 88.4% | 81.1% | 82.2% |
| | BRUMO25 | 93.2% | 93.3% | 93.3% | 93.3% | 93.3% | 93.3% | 93.3% |
| | HMMT25 | 69.6% | 79.5% | 69.9% | 69.9% | 83.3% | 69.9% | 69.9% |
| | GPQA | 72.5% | 70.6% | 71.2% | 72.2% | 71.4% | 72.3% | 72.5% |
| Qwen3-8B | AIME24 | 80.1% | 86.7% | 80.3% | 80.1% | 86.7% | 80.4% | 80.1% |
| | AIME25 | 82.6% | 76.4% | 82.7% | 82.7% | 74.3% | 82.6% | 82.7% |
| | BRUMO25 | 80.9% | 83.3% | 81.8% | 80.9% | 83.0% | 82.1% | 81.0% |
| | HMMT25 | 60.0% | 63.2% | 60.0% | 60.0% | 62.6% | 60.0% | 60.0% |
| | GPQA | 63.8% | 62.7% | 60.8% | 63.4% | 64.1% | 63.3% | 63.6% |
| Qwen3-32B | AIME24 | 85.3% | 90.8% | 86.0% | 85.5% | 92.8% | 86.2% | 85.7% |
| | AIME25 | 80.1% | 80.2% | 80.1% | 80.0% | 80.2% | 80.0% | 80.0% |
| | BRUMO25 | 93.3% | 93.3% | 93.3% | 93.3% | 93.3% | 93.3% | 93.3% |
| | HMMT25 | 63.3% | 63.3% | 63.2% | 63.4% | 62.4% | 63.3% | 63.3% |
| | GPQA | 72.2% | 70.0% | 70.0% | 72.3% | 72.5% | 71.9% | 72.3% |
| GPT-OSS-20B | AIME24 | 96.7% | 96.5% | 96.5% | 96.6% | 96.6% | 96.6% | 96.7% |
| | AIME25 | 95.3% | 95.0% | 95.2% | 95.3% | 94.1% | 95.1% | 95.3% |
| | BRUMO25 | 87.3% | 87.5% | 86.6% | 87.3% | 88.0% | 87.4% | 87.3% |
| | HMMT25 | 89.9% | 90.2% | 89.6% | 89.8% | 87.8% | 89.5% | 89.8% |
| GPT-OSS-120B | AIME24 | 96.7% | 96.5% | 96.3% | 96.6% | 96.6% | 96.6% | 96.6% |
| | AIME25 | 97.0% | 98.1% | 96.9% | 97.0% | 98.4% | 97.4% | 97.1% |
| | BRUMO25 | 86.7% | 82.9% | 85.3% | 86.4% | 85.1% | 86.7% | 86.7% |
| | HMMT25 | 92.9% | 90.5% | 92.9% | 92.9% | 84.8% | 93.0% | 92.9% |
| Average Acc. | | 83.0% | 84.0% | 82.6% | 83.0% | 84.0% | 83.0% | 83.0% |

configurations and 4 datasets (AIME24/AIME25/BRUMO25/HMMT25), totaling 20 experimental settings, and for each method we vary $B \in \{1, \ldots, 512\}$. The results are presented in Fig. 8.

**Keeping Top 90%** consistently matches or slightly outperforms unweighted majority voting with low variance ($-0.21\% \sim +0.73\%$, avg. $+0.17\%$).

**Keeping Top 10%** yields notable gains on most datasets (12/20 settings; $+0.26\% \sim +9.38\%$), with drops on the remaining eight ($-4.69\%$ to $-0.31\%$); the overall average improvement is $+1.22\%$. These regressions arise from rare cases where confidence concentrates on an incorrect answer ("confidently wrong").

Both settings substantially outperform the single-sample baseline ($B$=1; i.e., no voting): Top 90% delivers consistent margins (+5.83% $\sim$ +16.88%, avg. +10.57%), while Top 10% provides even larger gains (+5.26% $\sim$ +20.94%, avg. +11.62%).

Overall, Top 90% is a safe choice when stability is paramount, whereas Top 10% offers higher average performance with occasional regressions. These results demonstrate the reliability of the offline method using Lowest Group Confidence across diverse model scales and mathematical reasoning benchmarks. We additionally report results on GPQA-Diamond in Appendix F.

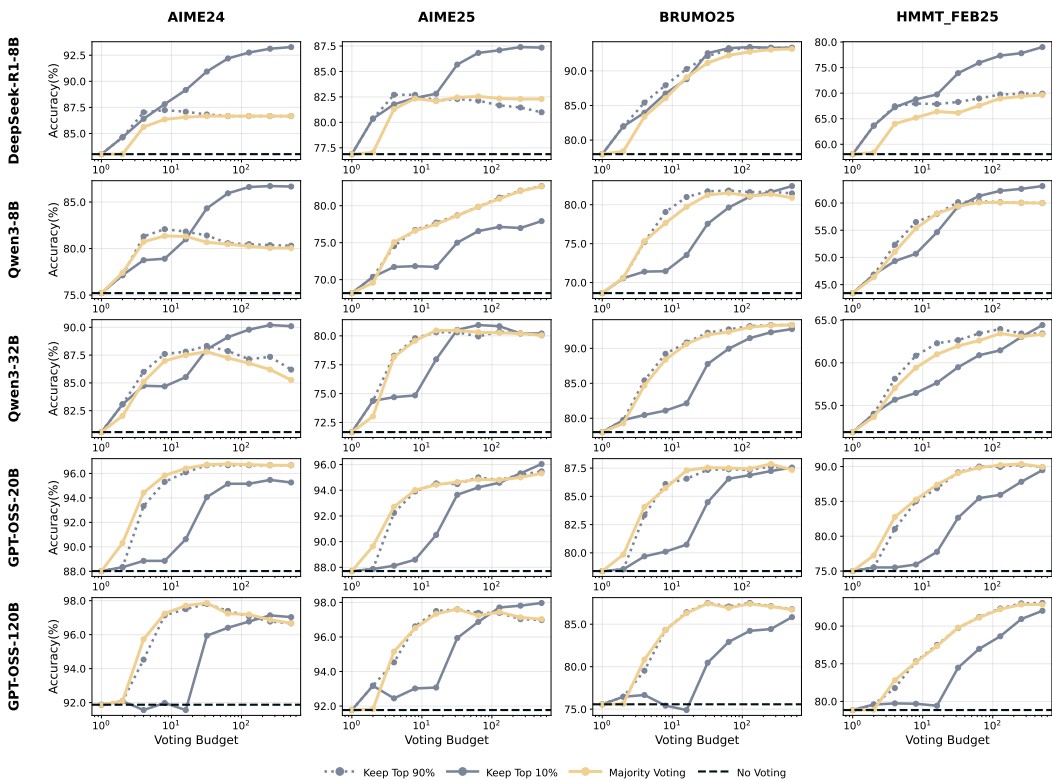

Figure 8: Scaling behavior: Models's Accuracy vs voting size for different methods on different models and datasets using offline method with Lowest Group Confidence

# E    SCALING BEHAVIOR FOR ONLINE DEEPCONF

We evaluate accuracy-cost trade-offs in an online setting in Fig. 9, where *cost* counts all generated tokens, including partially generated tokens from early-stopped traces. Each problem is warmuped with $N_{\text{init}}$=16 traces to *calibrate the consensus threshold $\tau$* (Sec. 3.3): We set $\tau$ to the 90th percentile for the DeepConf-low (top-10%) setting and to the 10th percentile for the DeepConf-high (top-90%) setting; a trace is stopped on the fly once its current group confidence falls below $\tau$. Aggregation over completed traces always uses confidence filtering plus confidence-weighted majority voting. We compare DeepConf with majority baseline at two perspectives.

**At matched budget.** We compare the adaptive DeepConf-high and DeepConf-low with majority voting at the voting budget of 512 in Table 10. Across models and datasets, DeepConf-low yields the largest cost reductions, which is about 43-84% fewer tokens than majority voting at $B$=512, while usually matching or improving accuracy (e.g., DeepSeek-8B/AIME24: $-77.9\%$ tokens, $+5.8$ pp; Qwen3-32B/AIME24: $-66.8\%$, $+4.7$ pp). DeepConf-high is more conservative, saving roughly 16-59% with accuracy essentially unchanged. Notable exceptions for DeepConf-low include Qwen3-8B/AIME25 ($-4.4$ pp) and a few <1 pp drops on GPT-OSS BRUMO/HMMT; on GPQA-Diamond, low still saves 55-65% with mixed (within $\pm 1.5$ pp) accuracy shifts. Overall, DeepConf-low offers

the best efficiency-accuracy trade-off, while DeepConf-high is the safer choice when minimizing accuracy changes is paramount.

Table 10: Benchmark DeepConf in the online setting. Accuracy (%) and tokens ($\times 10^8$) at voting size 512 for Majority Voting and Adaptive DeepConf-(high/low) on AIME24, AIME25, BRUMO25, HMMT25, GPQA-Diamond (where available; GPQA-Diamond only for DeepSeek/Qwen).

| Model | Dataset | Maj.@512 | | DeepConf-high@512 | | DeepConf-low@512 | |
|---|---|---|---|---|---|---|---|
| | | Tok | Acc | Tok ($\Delta\%$) | Acc | Tok ($\Delta\%$) | Acc |
| DeepSeek-8B | AIME24 | 3.55 | 86.7% | 1.45 (-59.0%) | 86.7% | 0.78 (-77.9%) | 92.5% |
| | AIME25 | 4.01 | 82.3% | 2.37 (-40.9%) | 81.4% | 1.24 (-69.0%) | 86.4% |
| | BRUMO25 | 3.56 | 93.3% | 2.17 (-39.2%) | 93.3% | 1.07 (-70.0%) | 93.3% |
| | HMMT25 | 4.49 | 69.8% | 3.43 (-23.5%) | 70.0% | 1.60 (-64.4%) | 77.6% |
| | GPQA-D | 9.92 | 72.5% | 6.90 (-30.4%) | 72.4% | 3.46 (-65.1%) | 71.7% |
| Qwen3-8B | AIME24 | 2.32 | 80.0% | 1.33 (-42.8%) | 80.4% | 0.90 (-61.1%) | 86.5% |
| | AIME25 | 2.77 | 82.5% | 1.99 (-28.1%) | 82.8% | 1.31 (-52.7%) | 78.1% |
| | BRUMO25 | 2.54 | 81.0% | 1.74 (-31.4%) | 81.7% | 1.15 (-54.7%) | 82.7% |
| | HMMT25 | 3.10 | 60.0% | 2.59 (-16.6%) | 60.0% | 1.67 (-46.2%) | 62.7% |
| | GPQA-D | 7.47 | 63.7% | 4.94 (-33.9%) | 63.8% | 3.31 (-55.7%) | 65.2% |
| Qwen3-32B | AIME24 | 2.00 | 84.8% | 0.88 (-56.0%) | 86.4% | 0.66 (-66.8%) | 89.5% |
| | AIME25 | 2.43 | 80.1% | 1.61 (-33.7%) | 80.2% | 1.14 (-52.9%) | 80.2% |
| | BRUMO25 | 2.17 | 93.3% | 1.37 (-37.1%) | 93.3% | 0.96 (-55.7%) | 92.4% |
| | HMMT25 | 2.76 | 63.4% | 2.24 (-18.8%) | 63.6% | 1.55 (-43.8%) | 64.5% |
| | GPQA-D | 7.44 | 72.2% | 4.16 (-44.1%) | 72.9% | 3.21 (-56.9%) | 73.0% |
| GPT-20B | AIME24 | 5.57 | 96.7% | 3.07 (-44.8%) | 96.7% | 1.11 (-80.0%) | 95.7% |
| | AIME25 | 6.26 | 95.4% | 3.18 (-49.2%) | 95.3% | 1.21 (-80.7%) | 96.1% |
| | BRUMO25 | 5.16 | 87.1% | 3.49 (-32.5%) | 87.2% | 1.34 (-74.1%) | 87.8% |
| | HMMT25 | 8.16 | 89.9% | 6.03 (-26.0%) | 90.3% | 2.17 (-73.4%) | 89.4% |
| GPT-120B | AIME24 | 2.66 | 96.7% | 1.20 (-54.6%) | 96.7% | 0.53 (-79.9%) | 97.0% |
| | AIME25 | 3.23 | 97.1% | 1.42 (-56.0%) | 97.0% | 0.49 (-84.7%) | 97.9% |
| | BRUMO25 | 2.68 | 83.8% | 1.81 (-32.6%) | 84.0% | 0.73 (-72.8%) | 83.4% |
| | HMMT25 | 4.09 | 92.8% | 2.78 (-32.0%) | 93.0% | 0.97 (-76.2%) | 92.0% |

**At Comparable Accuracy.** We compare adaptive DeepConf (early termination when the modal answer reaches $\geq 0.95$ confidence, otherwise continuing to the budget cap) against budget-only DeepConf (which always runs to the full budget cap) under two filtering regimes: High retains the top $90\%$ confidence traces, while Low retains only the top $10\%$. We conduct experiments across budget sizes $B \in \{32, 64, 128, 256, 512\}$ on AIME24/AIME25/BRUMO25/HMMT25 datasets (Fig. 9).

Benchmarked against majority voting baselines, adaptive DeepConf-low typically achieves 19–96% token reduction while maintaining matched accuracy, whereas adaptive DeepConf-high delivers 13–84% savings with near-equivalent performance. However, several exceptions exist within the $B \in [32, 512]$ range where matching majority voting accuracy with reduced token consumption is not achieved: DeepConf-low: Qwen3-8B/AIME25, Qwen3-8B/BRUMO25, Qwen3-32B/BRUMO25, GPT-20B/AIME24, GPT-20B/BRUMO25, GPT-20B/HMMT25, GPT-120B/BRUMO25, GPT-120B/HMMT25; DeepConf-high: DeepSeek-8B/AIME25.

Overall, DeepConf-low filtering provides the most substantial efficiency gains in successful cases, while DeepConf-high filtering represents the more conservative choice when minimizing accuracy degradation is critical. Compared to budget-only DeepConf, adaptive DeepConf consistently dominates the token–accuracy Pareto frontier: at identical voting ensemble sizes, it consumes fewer tokens without sacrificing accuracy (e.g., DeepSeek-8B/AIME24 @512: $0.782 \times 10^8$ vs $1.512 \times 10^8$ tokens for Low; GPT-120B/HMMT25 @512: $2.782 \times 10^8$ vs $3.679 \times 10^8$ for High). Consequently, we adopt adaptive DeepConf as the default configuration when computational efficiency is prioritized.

# F GPQA-DIAMOND RESULTS

We present the performance of our methods applied to DeepSeek-8B, Qwen3-8B, and Qwen3-32B on the GPQA-Diamond dataset in this section. The offline results using Lowest Group Confidence are shown in Fig. 10. Our method matches or exceeds the majority voting baseline in terms of peak accuracy. On Qwen3-8B and Qwen3-32B, Keeping Top-10% outperforms the baseline, while on DeepSeek-8B, keeping Top-90% roughly matches it and keeping Top-10% performs slightly lower.

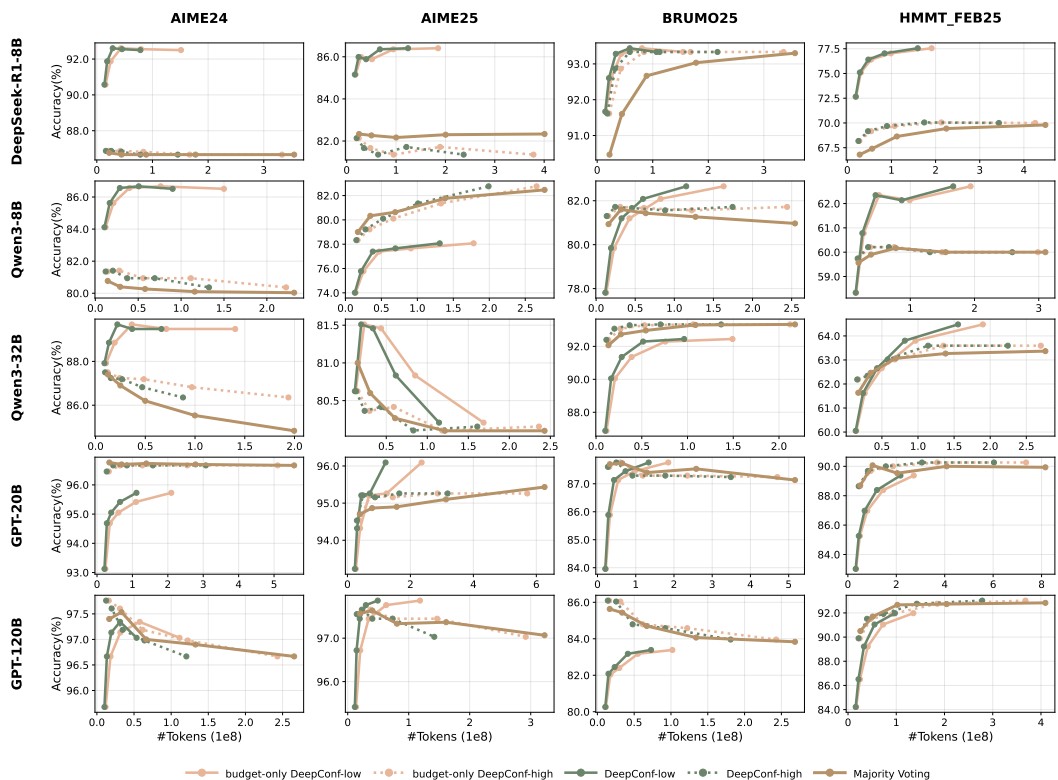

Figure 9: Scaling behavior: Models's Accuracy vs token cost for different methods on different models and datasets using online DeepConf

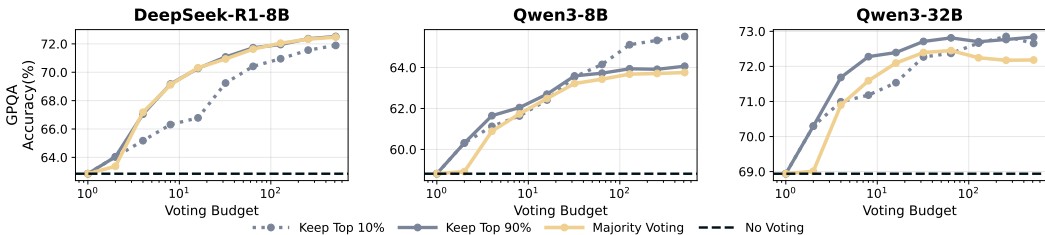

Figure 10: Scaling behavior: Models's Accuracy vs budget size for different methods on GPQA-Diamond

Overall, keeping Top-90% represents the safer choice, consistently matching or exceeding baseline accuracy, whereas keeping Top-10% often achieves larger gains but may occasionally underperform. Relative to generating only one answer per question, both variants provide clear average improvements of approximately 6%.

The online method's performance is shown in Fig. 11. Adaptive policies consistently achieve greater token usage reduction at the same voting budget compared to the fixed method. Consistent with the offline results, DeepConf-high generally maintains majority voting accuracy with a conservative approach, while DeepConf-low pursues larger computational savings but may underperform on DeepSeek-8B. These results align with the findings reported in §4.2 and §4.3.

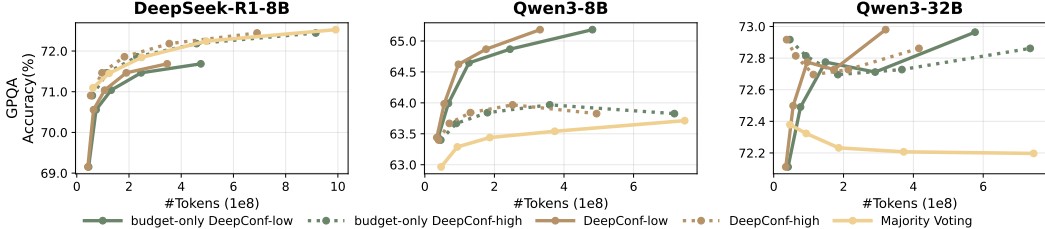

Figure 11: Scaling behavior: Models's Accuracy vs token cost for different methods on GPQA-Diamond

# G EXPERIMENTAL SETTINGS

***Online DeepConf hyperparameters.*** Table 12 summarizes the settings used in our runs ( $N_{\text{init}}$, $\eta$, $\tau$, and the voting budget $B$ ), with two configurations: *DeepConf-low* ($\eta$=10%) and *DeepConf-high* ($\eta$=90%).

***Generation hyperparameters.*** We list below the per-model decoding hyperparameters used across all experiments. For each model, we fix temperature, top-$p$, top-$k$, and the maximum generation length, and we use each model's native tokenizer. A dash (—) indicates that the control is not applied (e.g., if top-$k$ is —, sampling uses only top-$p$ truncation).

Table 11: Generation hyperparameters used in our experiments. Different models use different decoding settings. A dash (—) indicates the option is not applied.

| Model | Temperature | Top-$p$ | Top-$k$ | Max seq len |
|---|---|---|---|---|
| DeepSeek-8B | 0.6 | 0.95 | — | 64k |
| Qwen3-8B | 0.6 | 0.95 | 20 | 32k |
| Qwen3-32B | 0.6 | 0.95 | 20 | 32k |
| GPT-OSS-20B | 1.0 | 1.0 | 40 | 130k |
| GPT-OSS-120B | 1.0 | 1.0 | 40 | 130k |

***Prompt templates.*** For `Qwen3` and `GPT-OSS`, we append the same instruction to every problem prompt: "Please reason step by step, and put your final answer within \boxed{}." For `GPT-OSS`, we additionally keep the provider's official system prompt and enable the *reasoning effort = high* setting. For `DeepSeek-8B`, we use the official system prompt and put the problem in the user message.

In all cases, the final answer is expected to appear inside \boxed{...} and is extracted during post-processing. Decoding terminates only when an end-of-sequence token is produced or the maximum generation length is reached.

Table 12: Hyperparameters for *Online DeepConf* (Algorithm 2). $N_{\text{init}}$ denotes the number of initial traces used in the offline warmup; $\eta$ is the filtering percentile to form $T_{\text{top}}$ (we will keep top $\eta$ traces); $\tau$ is the online consensus threshold; $B$ is the maximum budget (number of traces).

| Method | $N_{\text{init}}$ | $\eta$ (Top-%) | $\tau$ (consensus) | $B$ (budget) |
|---|---|---|---|---|
| DeepConf-low | 16 | 10% | 0.95 | $32, 64, 128, 256, 512$ |
| DeepConf-high | 16 | 90% | 0.95 | $32, 64, 128, 256, 512$ |

