# OpenReview forum: "Deep Think with Confidence"
_ICLR.cc/2026/Conference — ICLR 2026 Poster_

### Official Review · Reviewer_2dPu · 2025-10-29

**Soundness:** 3
**Presentation:** 2
**Contribution:** 3
**Rating:** 4
**Confidence:** 4

**Summary:**

This paper introduces Deep Think with Confidence (DeepConf), a test-time method to enhance the reasoning performance and efficiency of Large Language Models (LLMs) by leveraging internal confidence signals from token distributions. DeepConf filters low-quality reasoning traces either offline (after generation) or online (during generation) using localized confidence measures like group confidence, bottom 10% group confidence, lowest group confidence, and tail confidence. It integrates with self-consistency majority voting, achieving weighted voting and early stopping to reduce computational overhead. Evaluations on mathematical benchmarks (e.g., AIME 2025) and models (e.g., GPT-OSS-120B, Qwen3-32B) claim up to 99.9% accuracy with 84.7% token reduction compared to standard majority voting. The method requires no additional training and is plug-and-play.

**Strengths:**

-  DeepConf demonstrates significant reductions in generated tokens (up to 84.7%) while maintaining or improving accuracy, which is crucial for deploying LLMs in resource-constrained settings. The online mode with adaptive sampling and warmup is a clever way to approximate offline performance in real-time, making it suitable for serving frameworks.
- Experiments cover multiple models (from 8B to 120B parameters) and challenging benchmarks (AIME, HMMT, BRUMO, GPQA). Results are averaged over 64 repetitions, adding statistical robustness. The ablation on confidence thresholds (η=10% vs. 90%) provides insights into trade-offs between precision and diversity in filtering.
-  No hyperparameter tuning or retraining is needed, and the method integrates seamlessly with existing parallel thinking approaches. The anonymous code release supports reproducibility.

**Weaknesses:**

- The paper assumes that higher confidence correlates with correctness (as shown in Figure 2), but this assumption is problematic. Confidence in large models is not necessarily calibrated, and without calibration analysis (e.g., ECE or reliability plots), confidence values cannot be interpreted as correctness probabilities. Moreover, low-confidence traces may still contain valid, diverse answers that are filtered out, potentially causing mode collapse. A more thorough analysis of calibration quality and failure modes (e.g., overconfident wrong answers) would be necessary to substantiate the claims.
-  While the idea of discarding uncertain reasoning paths during generation makes sense conceptually, the experimental setting (K=512) raises concerns about practicality. The paper does not report the computational overhead or the average inference cost compared to normal decoding. It is unclear how scalable the method is in realistic settings and how strongly the results depend on the choice of K. An ablation or sensitivity analysis on K would be essential to understand the trade-off between accuracy and efficiency.
- The paper mainly evaluates the method on benchmarks with well-defined ground truth answers. To better assess its generalization ability, it would be helpful to include experiments on open-ended tasks where correctness is less strictly defined. Such evaluation could reveal whether the proposed confidence mechanism remains effective under less constrained, real-world generation scenarios.
If the authors can address the above issues, I would be willing to raise my score.

**Questions:**

In addition to the weaknesses mentioned above, I am particularly concerned about the practical computational cost of DeepConf. The reported results are obtained under the setting of K = 512, which seems computationally heavy. It would be important to clarify the average inference cost and compare it with standard decoding. Moreover, how sensitive is the method to the choice of K? How does the performance change when K is reduced to a more realistic range for deployment?

---

> ### Author Response · Authors · 2025-11-25
>
> We thank the reviewer for the detailed and thoughtful feedback. Below we address the main concerns regarding calibration, computational cost, and generalization.
>
> **W1:** The paper assumes that higher confidence correlates with correctness (as shown in Figure 2), but this assumption is problematic. Confidence in large models is not necessarily calibrated, and without calibration analysis (e.g., ECE or reliability plots), confidence values cannot be interpreted as correctness probabilities. Moreover, low-confidence traces may still contain valid, diverse answers that are filtered out, potentially causing mode collapse. A more thorough analysis of calibration quality and failure modes (e.g., overconfident wrong answers) would be necessary to substantiate the claims.
>
> **A1:** We agree that calibration is a critical factor. To verify the reliability of our confidence signals, we conducted a rigorous analysis of AUROC and ECE on AIME24 and HMMT (Feb 2025) using Qwen3-32B.
>
> Regarding the risk of filtering valid answers: Our metrics show an AUROC > 0.90 (Table below). This high discriminative power indicates that low-confidence scores are effective indicators of incorrectness, suggesting that valid traces are rarely filtered, thereby mitigating the concern of mode collapse.
>
> Regarding calibration: We found that our confidence signals are well-calibrated after standard alignment. Since raw scores are not probabilities, we applied Platt Scaling (Logistic Regression) to map them. The resulting Expected Calibration Error (ECE) is < 0.05. These results demonstrate that the model's internal confidence can serve as a reliable proxy for correctness.
>
> Table: Quantitative Evaluation of Confidence Metrics (AUROC & ECE)
>
>  | Dataset | Metric | AUROC ($\uparrow$) | ECE (Platt) ($\downarrow$) |
> | :-- | :-- | :-- | :-- |
> | AIME24 | Mean Confidence | 0.9151 | 0.0371 |
> | | Bottom 0.1  | 0.9088 | 0.0253 |
> | | Tail 2048 | 0.9006 | 0.0442 |
> | HMMT (Feb 25) | Mean Confidence | 0.9220 | 0.0346 |
>  | | Bottom 0.1  | 0.9148 | 0.0300 |
>  | | Tail 2048 | 0.9052 | 0.0195 |
>
> Note: ECE is reported after 5-fold cross-validated Platt scaling. This standard step is necessary to map our raw, non-probabilistic confidence scores into probabilities for a valid calibration analysis.
>
> **W2&Q1:** While the idea of discarding uncertain reasoning paths during generation makes sense conceptually, the experimental setting (K=512) raises concerns about practicality. The paper does not report the computational overhead or the average inference cost compared to normal decoding. It is unclear how scalable the method is in realistic settings and how strongly the results depend on the choice of K. An ablation or sensitivity analysis on K would be essential to understand the trade-off between accuracy and efficiency. In addition to the weaknesses mentioned above, I am particularly concerned about the practical computational cost of DeepConf. The reported results are obtained under the setting of K = 512, which seems computationally heavy. It would be important to clarify the average inference cost and compare it with standard decoding. Moreover, how sensitive is the method to the choice of K? How does the performance change when K is reduced to a more realistic range for deployment?
>
> **A2:** We clarify that our primary baseline is Self-Consistency (SC), which has become the industrial standard for scaling test-time compute (e.g., similar to strategies used in Grok’s heavy mode). Within this context, DeepConf is designed to make high-sample regimes feasible, not to compete with single-path decoding.
>
> While K=512 seems heavy, it is often necessary for peak reasoning performance, and DeepConf makes it economically feasible. By pruning unpromising paths early, we reduce total token consumption by up to 84.7%. This means running DeepConf at K=512 incurs roughly the same inference cost as standard SC at K=80. Besides, the overhead for computing confidence is virtually zero; calculating confidence is just simple arithmetic on logprobs, which is negligible compared to the heavy matrix multiplications required to generate tokens.
>
> Regarding sensitivity to K, Figure 5/7/8/9/10 already provides this analysis by plotting performance across budgets corresponding to K = {32, 64, 128, 256, 512}. The results show that DeepConf maintains its efficiency advantage across the entire range. Crucially, even at a practical low-K setting (e.g., K=32), our method outperforms single-path decoding. This confirms that DeepConf is not limited to heavy reasoning modes; it is also a scalable and effective solution for realistic deployment.

---

> > ### Author Response · Authors · 2025-11-25
> >
> > **W3:** The paper mainly evaluates the method on benchmarks with well-defined ground truth answers. To better assess its generalization ability, it would be helpful to include experiments on open-ended tasks where correctness is less strictly defined. Such evaluation could reveal whether the proposed confidence mechanism remains effective under less constrained, real-world generation scenarios. If the authors can address the above issues, I would be willing to raise my score.
> >
> >
> > **A3:** We agree that evaluating generalization on open-ended tasks is a valuable direction. Our current work prioritizes well-grounded reasoning tasks with fixed-form answers, as these allow for rigorous, quantitative evaluation via standard majority voting.
> >
> > However, we clarify that DeepConf is conceptually compatible with open-ended generation. As demonstrated by Universal Self-Consistency [1], self-consistency can be extended to free-form text by aggregating responses (e.g., using an LLM to judge semantic equivalence). DeepConf naturally enhances this framework: instead of treating all generated drafts equally, our confidence scores can serve as reliability weights for the 'LLM-as-a-judge.' For instance, the judge could prioritize clusters backed by high-confidence traces. While implementing this pipeline requires a distinct evaluation setup, we consider it a promising direction for future work.
> >
> > [1] Chen, X., Aksitov, R., Alon, U., Ren, J., Xiao, K., Yin, P., ... & Zhou, D. (2023). Universal self-consistency for large language model generation. arXiv preprint arXiv:2311.17311.

---

> > > ### Comment · Reviewer_2dPu · 2025-11-27
> > >
> > > We thank the authors for their explanations and for addressing some of our previous concerns. However, I remain some concerns regarding W1 and W3, specifically:
> > >
> > > W1: Reliability of confidence signals.
> > >
> > > Although the authors conducted additional experiments in the rebuttal stage to verify the reliability of the confidence signals used, I remain skeptical about the reported results. In Table 1, nearly all AUROC values are above 0.90 and ECE values are below 0.05. However, prior work has shown that using logits as confidence scores is often unreliable [1][2]. The results reported here seem overly optimistic. The authors should provide a more detailed explanation of why their logits perform so well as confidence scores. Is this behavior related to the model type, the specific tasks or data, or other factors?
> > >
> > > W3: Applicability of DeepConf in open-domain tasks.
> > >
> > > This concern is closely related to W1. If the logits-based confidence scores are indeed reliable, then DeepConf could be applied directly to open-domain tasks. In this case, the authors should add the experimental results on open-domain tasks to further validate this method. Conversely, if logits are not reliable, the authors should clarify alternative signals to trigger the regeneration process.
> > >
> > > Overall, I believe these two points are critical for this paper. The authors should provide a more thorough discussion or additional experimental evidence addressing these concerns.
> > >
> > > [1] Shifting Attention to Relevance: Towards the Predictive Uncertainty Quantification of Free-Form Large Language Models. ACL, 2024
> > >
> > > [2] When to Trust LLMs: Aligning Confidence with Response Quality. ACL, 2024

---

> > > > ### Author Response · Authors · 2025-11-29
> > > >
> > > > We thank the reviewer for their continued engagement and for acknowledging our clarifications.
> > > >
> > > > **W1:** Reliability of confidence signals.
> > > >
> > > > **R1:** We respectfully note that the cited works ([1][2]) primarily analyze older model generations. Recent benchmarks [3] demonstrate that SOTA post-trained reasoning models (like the Qwen3 used here) exhibit significantly improved calibration. Crucially, Figure 2 in [3] explicitly highlights that token-level signals in reasoning models are particularly strong indicators of correctness, directly supporting our use of token-level aggregation. This aligns with the intrinsic nature of the domain: unlike ambiguous open-domain QA, math and logic problems produce sharper signals because a logical step is strictly valid or invalid. This binary characteristic makes confidence a far more discriminative proxy for correctness than in general knowledge tasks.
> > > >
> > > > **W3:** Applicability of DeepConf in open-domain tasks.
> > > >
> > > > **R3:** We maintain that open-domain evaluation falls outside the scope of this work. The standard evaluation paradigm for reasoning models centers on closed-ended tasks (e.g., Math, Code) precisely because they offer objective ground truth, whereas open-ended tasks introduce significant evaluator noise. We believe our extensive validation across multiple benchmarks and model scales already constitutes a complete and rigorous study. Furthermore, adapting DeepConf to subjective domains requires non-trivial effort (e.g., calibrating confidence against preference models rather than logical exact-match), which represents a distinct research challenge. We therefore reserve this extension for future work.
> > > >
> > > > [1] Duan, Jinhao, et al. "Shifting attention to relevance: Towards the predictive uncertainty quantification of free-form large language models." Proceedings of the 62nd Annual Meeting of the Association for Computational Linguistics (Volume 1: Long Papers). 2024.
> > > >
> > > > [2] Tao, Shuchang, et al. "When to trust llms: Aligning confidence with response quality." arXiv preprint arXiv:2404.17287 (2024).
> > > >
> > > > [3] Tao, Linwei, et al. "Revisiting Uncertainty Estimation and Calibration of Large Language Models." arXiv preprint arXiv:2505.23854 (2025).

---

### Official Review · Reviewer_yu1z · 2025-10-30

**Soundness:** 3
**Presentation:** 3
**Contribution:** 2
**Rating:** 4
**Confidence:** 4

**Summary:**

Self-consistency with majority voting as a test-time scaling (TTS) strategy can effectively improve LLM reasoning accuracy; however, it multiplies computational cost and shows pronounced diminishing returns. To address this, the paper proposes DeepConf, an efficient, training-free TTS method that leverages a model’s internal confidence to dynamically filter low-quality reasoning paths. Across both offline and online settings, DeepConf achieves superior performance while using substantially fewer resources.

**Strengths:**

1. **Simple and effective with practical value.** Relying only on *Bottom 10% Group Confidence* enables effective adaptive sampling with early stopping and confidence filtering of high-quality reasoning paths. In both offline and online settings, it markedly lowers inference cost while delivering better performance, indicating broad applicability.

2. **Comprehensive and rigorous experiments.** The paper evaluates five mainstream open-source reasoning LLMs from three model families on five challenging benchmarks under two settings. Experimental details are clear and easy to follow. Thorough ablations, across different budgets and hyperparameters--demonstrate the method’s effectiveness, making the evidence methodologically solid.

3. **Clear writing and sound structure.** Centered on confidence and targeting cost-efficient TTS, the paper offers a detailed, comprehensive discussion and comparison of approaches based on internal token distributions and their limitations, and builds a logically coherent, step-by-step case for the proposed method.

**Weaknesses:**

1. **Limited task generalization.** The evaluation focuses primarily on mathematical reasoning, with little assessment in verifiable settings such as code generation. Adding at least one non-mathematical reasoning benchmark would strengthen claims of generality.

2. **Missing highly relevant strong baselines.** While the paper centers on confidence-based methods and compares unweighted vs. confidence-weighted majority voting, it lacks comparisons with closely related, strong baselines—e.g., ESC [1], DSC [2], and RASC [3]. Including these would improve completeness, better substantiate the method’s effectiveness, and clarify the source of gains.

3. **Missing related work** . The paper omits several classic/recent SOTA parallel-generation approaches under TTS in A.1, such as DVTS [4] and DORA [5].

[1] Escape Sky-high Cost: Early-stopping Self-Consistency for Multi-step Reasoning ICLR2024

[2] Make Every Penny Count: Difficulty-Adaptive Self-Consistency for Cost-Efficient Reasoning NAACL2025

[3] Leveraging Reasoning Paths for Efficient LLM Sampling NAACL2025

[4] Scaling Test-Time Compute with Open Models

[5] Every Rollout Counts: Optimal Resource Allocation for Efficient Test-Time Scaling NeurIPS2025

**Questions:**

1. **Online, extreme-difficulty cases.** When the model generates a long chain-of-thought (CoT), if certain “reflection” segments exhibit low confidence, could early stopping become overly aggressive—filtering out trajectories that might otherwise converge to the correct answer—and thus hurt accuracy? Should the definition of the group (G_i) adapt to the already-produced output length?

2. **Practical guidance on hyperparameters.** What are the recommended defaults and applicable ranges for (G_i), (\eta), (\tau), and (N_{\text{init}})? Is there a simple heuristic or automatic setting strategy for these?

---

> ### Author Response · Authors · 2025-11-25
>
> We thank the reviewer for the thoughtful comments and for recognizing the practical value, strong experimental design, and clarity of our paper. We address the key concerns below.
>
> **W1:** Limited task generalization. The evaluation focuses primarily on mathematical reasoning, with little assessment in verifiable settings such as code generation. Adding at least one non-mathematical reasoning benchmark would strengthen claims of generality.
>
> We agree that broader task coverage is valuable. However, we respectfully remind the reviewer that our evaluation already includes **GPQA-Diamond**, which extends beyond mathematics to cover graduate-level physics, biology, and computer science.
>
> Our results on this benchmark confirm that DeepConf generalizes well to non-mathematical reasoning. Specifically, DeepConf-high reliably saves 30–45% of generation, while DeepConf-low reduces token usage by 55–65%. This demonstrates that our method is effective across diverse domains, not just in pure math settings.
>
> **W2:** Missing Baselines: While the paper centers on confidence-based methods and compares unweighted vs. confidence-weighted majority voting, it lacks comparisons with closely related, strong baselines—e.g., ESC [1], DSC [2], and RASC [3]. Including these would improve completeness, better substantiate the method’s effectiveness, and clarify the source of gains.
>
> We have added comparisons with ESC (ICLR’24) and DSC (NAACL’25). Consistent with their design goals, both methods mainly reduce token cost while leaving accuracy essentially unchanged from standard SC. In contrast, DeepConf improves both accuracy and efficiency. We compare the results of Qwen3-32B on easier AIME24 and harder HMMT25 at K=512.
>
> On AIME24, DeepConf reaches 89.48% at K=512, a +4.22pp gain over SC and +4.32pp over DSC, while using 66.24M tokens for the whole dataset, which is 66.8% fewer than SC and 13.6% fewer than DSC. On HMMT, DeepConf is again strictly better: at K=512, it improves accuracy to 64.48% (+1.15pp) while reducing tokens by 43.8% vs SC. These results show our gains are not just from early stopping, but from confidence-guided pruning/weighting that makes SC both cheaper and more accurate.
>
> | Dataset | K | Method | Acc (%) | Tokens (M) | Savings vs. SC (%) |
> |:--|--:|:--|--:|--:|--:|
> | **AIME24** | **512** | SC | 85.26 | 199.63 | 0.0 |
> |  |  | ESC warmup=8  | 85.21 | 107.75 | 46.0 |
> |  |  | ESC warmup=16  | 85.21 | 132.43 | 33.7 |
> |  |  | DSC | 85.16 | 76.70 | 61.6 |
> |  |  | **DeepConf (ours)** | **89.48** | **66.24** | **66.8** |
> | **HMMT** |**512** | SC  | 63.33 | 276.40 | 0.0 |
> |  |  | ESC warmup=8  | 63.33 | 234.97 | 15.0 |
> |  |  | ESC warmup=16  | 63.33 | 248.73 | 10.0 |
> |  |  | DSC | 63.33 | 214.90 | 22.3 |
> |  |  | **DeepConf (ours)** | **64.48** | **155.41** | **43.8** |
>
>
> **W3:** Missing Related Work - The paper omits several classic/recent SOTA parallel-generation approaches under TTS in A.1, such as DVTS [4] and DORA [5].
>
> We appreciate this suggestion. We have expanded Section A of the Related Work to include recent parallel generation and TTS methods and we have updated the paper PDF. These additions will better situate DeepConf within the landscape of cost-efficient test-time reasoning strategies.
>
> **Q1:** Online, extreme-difficulty cases. When the model generates a long chain-of-thought (CoT), if certain “reflection” segments exhibit low confidence, could early stopping become overly aggressive—filtering out trajectories that might otherwise converge to the correct answer—and thus hurt accuracy? Should the definition of the group (G_i) adapt to the already-produced output length?
>
> We acknowledge the risk of pruning valid self-corrections, but our large sliding window (2048 tokens) is designed specifically to mitigate this. It smooths out short, transitional dips in confidence (like "wait, let me re-think"), ensuring we only target sustained reasoning failures rather than momentary hesitations.
>
> Regarding group sizing: we currently find fixed windows offer the best stability, but we agree that adapting window size to generation length is a promising direction for future work.

---

> > ### Author Response · Authors · 2025-11-25
> >
> > **Q2:** Practical guidance on hyperparameters. What are the recommended defaults and applicable ranges for (G_i), (\eta), (\tau), and (N_{\text{init}})? Is there a simple heuristic or automatic setting strategy for these?
> >
> > We already provide default values and settings in the main text and Appendix G. To summarize:
> > - $G_i$: group size = 2048 tokens (fixed)
> > - $\eta$: retention ratio = 10% (aggressive) or 90% (conservative)
> > - $\tau$: consensus threshold = 0.95 (fixed)
> > - $N_{\text{init}}$: warmup size = 16 (fixed)
> >
> > These values were chosen based on general properties of LLM output distributions (e.g., typical reasoning step lengths and confidence concentration). While our ablation studies (Table 5/Fig. 8) demonstrate that performance can be marginally optimized by tuning these values, the core results show that our default setting is robust and performs consistently strong across diverse tasks and models.

---

> ### Comment · Reviewer_yu1z · 2025-11-26
>
> Thank you for your detailed response. Since my concerns are addressed, I decide to raise my score to 6. I hope the authors can update their paper to reflect the above changes.

---

### Official Review · Reviewer_s7Pe · 2025-10-31

**Soundness:** 3
**Presentation:** 3
**Contribution:** 3
**Rating:** 6
**Confidence:** 2

**Summary:**

This paper introduces Deep Think with Confidence (DeepConf), a test-time method designed to improve the reasoning performance of LLMs. The paper argues that global confidence metrics are flawed as few high confident region can hide critical errors within low confidence segment. It proposes several new, more fine grained metrics such as group confidence and tail confidence, etc to capture local statistics. It then introduces two inference mode, in offline settings confidence filtering and weighted majority voting is applied. In online settings, an offline warmup is required to setup a stopping threshold, and adaptive sampling can be applied based on existing rollout results and partial confidence. The method demonstrates a accuracy improvement compared with majority voting and significant token saving in online settings.

In short, it is an empirical paper that extends the previous global confidence method with finer granularity to improve the test-time performance on token efficiency and accuracy.

**Strengths:**

1. Clear Motivation and Problem Framing: The paper provides a clear and intuitive justification for moving from global to local confidence metrics.
2. The experimental setup is comprehensive: The authors test across multiple model families and scales, on a suite of reasoning and QA benchmarks. The paper also includes detailed ablations on hyperparameters introduced in the method.
3. The proposed method is practical and simple with multiple variants that can adapte to different downstream need, making it easy and meaningful to integrate into existing frameworks.

**Weaknesses:**

1. Incremental Novelty: The primary novelty lies in the specific local metrics and the online early-stopping mechanism compared to prior works (e.g., Kang et al., 2025). While effective, the proposed method is an incremental improvement on existing ideas.
2. Overstated “No Hyperparameter Tuning.” The abstract’s “no … hyperparameter tuning” assertion is not faithful to the method as presented. The method introduces several new parameters that require choices: the filtering ratio, the online consensus threshold, the warmup steps and window sizes. For example, the reported ablations (e.g., Table 5 / Fig. 8) show the optimal filtering percentage varies by dataset and model, implying some form of tuning or at least selection is needed for new settings.
3. Warmup Cost Not Fully Accounted. The online mode requires an instance-level warmup that generates N full traces before early stopping is even applicable. For common majority-vote budgets e.g. K=32, setting N=16 is a major overhead that should be counted. The current amortized token-saving curves appear to exclude or underweight this cost.

**Questions:**

1. As for weakness 2: What default, model-agnostic settings do you recommend for these parameters? What minimal selection protocol (and validation budget) do you propose for new tasks/models? if above questions cannot be addressed, please revise the abstract/claims to acknowledge required hyperparameter choices.
2. Regarding Weakness 3: Can the authors provide the token efficiency metrics taking offline warmup tokens into account? Can the confidence threshold be reused or shared across questions or within a dataset to reduce per-prompt warmup?

---

> ### Author Response · Authors · 2025-11-25
>
> We thank the reviewer for the detailed feedback and for recognizing the motivation, broad applicability, and comprehensive evaluation of DeepConf. We respond to the main concerns below.
>
> **W1:** Incremental Novelty: The primary novelty lies in the specific local metrics and the online early-stopping mechanism compared to prior works (e.g., Kang et al., 2025). While effective, the proposed method is an incremental improvement on existing ideas.
>
> We respectfully clarify that moving from global, offline filtering to local, online early-stopping is a key difference, not just a small update. Unlike Kang et al. (2025), which relies on global confidence from the entire output, DeepConf introduces local confidence patterns, such as Lowest Group confidence, that are computed incrementally during decoding. This allows us to terminate individual traces immediately when local uncertainty is high. This localized and adaptive mechanism enables a novel online algorithm for self-consistency reasoning that is both efficient and model-agnostic, which is something offline methods strictly cannot support.
>
> This mechanism yields **non-trivial** efficiency gains that prior offline methods cannot achieve. Empirically, on challenging benchmarks like AIME 2025, our offline evaluation confirms the high reliability of the local confidence signal (achieving up to 99.9% accuracy). Crucially, the proposed online mechanism effectively leverages this signal to reduce generated tokens by up to 84.7%. These order-of-magnitude improvements in computational efficiency demonstrate that the proposed DeepConf offers a distinct and substantial advancement over existing confidence-based baselines.
>
> **W2 & Q1:** Hyperparameter Tuning and Default Settings
>
> We clarify that we used a fixed default configuration (Window=2048, Warmup=16, Consensus=95%) across all main experiments without dataset-specific tuning. We already provide default values and settings in the main text and Appendix G. For the filtering ratio, we provide two standard presets: a conservative 90% (guarantees performance better than or as good as majority voting) and an aggressive 10% (maximizes efficiency gains).
>
> These values were chosen based on general properties of LLM output distributions (e.g., typical reasoning step lengths and confidence concentration). While our ablation studies (Table 5/Fig. 8) demonstrate that performance can be marginally optimized by tuning these values, the core results show that our default setting is robust and performs exceptionally well across diverse tasks and models.
>
> Thus, we believe the claim holds in practice: DeepConf works "out of the box" with these defaults. However, to ensure maximum precision and avoid ambiguity, we will revise the abstract to state that the method requires "no task-specific tuning", distinguishing it from strictly "parameter-free" approaches.
>
> **W3 & Q2.1:** Can the authors provide the token efficiency metrics taking offline warmup tokens into account? Can the confidence threshold be reused or shared across questions or within a dataset to reduce per-prompt warmup?
>
> All our online token-efficiency plots (e.g., Fig. 7) already include warm-up cost. Specifically, for each voting budget K (e.g., 32, 64, …), we allocate a fixed warmup size of 16, and then sample until reaching K traces total (i.e., 32=16 + 16, 64=16 + 48, …). The token usage of the online algorithm reported in the paper already includes: 1) All tokens from the warm-up traces, 2) All tokens from fully sampled traces after warm-up, 3) And early-stopped traces, counted up to their exit point.
>
> Thus, the token cost already reflects all computation, including the initial warm-up. We will clarify this in the caption and main text for completeness.
>
> **Q2.2:** Can the confidence threshold be reused or shared across questions or within a dataset to reduce per-prompt warmup?
>
> No, fixed thresholds cannot be reused to reduce per-prompt warmup. Confidence distributions vary too much with problem difficulty. For example, on all questions of AIME 2025, the top-10% threshold for Qwen3-8B ranged from 11.5 to 15.5. A single static number fails here. It ends up being too loose for easy questions and too strict for hard ones. Therefore, filtering must be query-adaptive, adjusting dynamically to the current input rather than relying on a fixed global value, even for problems from the same domain.

---

### Official Review · Reviewer_U1LE · 2025-11-01

**Soundness:** 3
**Presentation:** 4
**Contribution:** 3
**Rating:** 8
**Confidence:** 3

**Summary:**

This paper presents Deep Think with Confidence (DeepConf), a test-time method that improves LLM reasoning accuracy and efficiency by leveraging internal confidence signals without retraining. The method addresses the inefficiency and diminishing returns of standard self-consistency (majority voting) by dynamically filtering or halting low-confidence reasoning traces. DeepConf replaces a single global confidence value with localized confidence metrics computed over sliding windows of tokens:
- Group confidence: mean of token-level confidence across overlapping windows;
- Tail / Bottom-10% confidence: highlight locally uncertain regions;
- Lowest Group Confidence: the weakest confidence window per trace, used for online early stopping.

In offline mode, DeepConf ranks and filters traces by these scores, applying confidence-weighted voting instead of naive majority voting.
In online mode, it adaptively terminates traces early once local confidence falls below a learned percentile threshold and stops sampling once vote consensus exceeds a preset $\tau$.

Across math-reasoning and QA benchmarks (AIME’24/’25, HMMT’25, BRUMO’25, GPQA-Diamond) and multiple models (DeepSeek-8B, Qwen3-8B/32B, GPT-OSS-20B/120B), the method achieves strong empirical results without any retraining or extra supervision.

**Strengths:**

1. Practical impact and simplicity: DeepConf can be deployed immediately without retraining or external calibration. It’s a drop-in improvement to any self-consistency pipeline.
2. Local vs. global insight: The authors convincingly show that global average confidence can mask local failures while sliding-window metrics like LGC and Tail Confidence capture these weak spots better.
3. Strong empirical results: Large-scale experiments (5 datasets, 4 models, 64× repeats) demonstrate consistent improvements in both accuracy and efficiency.
4. Comprehensive ablations: The authors systematically explore the effects of confidence thresholding, consensus level, and warm-up size, providing actionable insights for practitioners.
5. Presentation: The presentation, visualizations, and overall narrative are coherent and reader-friendly.

**Weaknesses:**

1. Confidence calibration: The authors treat percentile thresholds ($\eta = 10 / 90$) as hyperparameters, but the actual confidence scores are not calibrated across models. Reporting AUROC or ECE/Brier scores for trace-level discrimination would provide a more rigorous evaluation of confidence quality.
2. “Confidently wrong” cases: The authors briefly mention (Appendix D) that models sometimes assign high confidence to incorrect reasoning modes, but there’s no robust mitigation strategy. A hybrid ensemble or diversity-aware weighting could improve robustness.
3. Hyperparameters: The choice of top-$k$ = 20 and window = 2048 seems arbitrary. An additional ablation varying these values could offer deeper understanding of sensitivity.

**Questions:**

1. How does DeepConf compare to semantic entropy and self-certainty in online settings under the same token budget?
2. Can the authors provide quantitative measures (e.g., AUROC, ECE) showing how well their confidence metrics discriminate correct vs. incorrect traces?
3. How often do “confidently wrong” traces dominate the vote, and can local diversity or disagreement be incorporated as a second signal?
4. Can the same percentile thresholds $s$ learned during warm-up be reused across multiple problems from the same domain, or must they be recomputed for every new query?
5. Would varying the top-k (for confidence) or window size meaningfully affect results?

---

> ### Author Response · Authors · 2025-11-25
>
> We thank the reviewer for the detailed and positive assessment of DeepConf. We address the main questions and weaknesses below.
>
> **Q1:** How does DeepConf compare to semantic entropy and self-certainty in online settings under the same token budget?
>
> We have added online comparisons under matched average token budgets with semantic-entropy based stopping and standard SC.
> We added online comparisons under the same max-K token budget with semantic-entropy stopping (Semantic-Entropy of threshold 0.8 and Semantic-Entropy of threshold 0.5) and standard self-consistency (SC). Semantic-entropy methods mainly reduce tokens but do not improve accuracy over SC at the same budget. Moreover, because they rely on post-hoc semantic clustering/entropy estimation, they cannot dynamically stop an ongoing request during generation in online serving.
> On HMMT using Qwen3-32B, DeepConf outperforms both SC and SE: it achieves higher accuracy while using fewer tokens than SC under K=128 and K=512 budgets, whereas semantic-entropy yields cost savings without gains.
>
> | Dataset | K | Method | Acc (%) | Tokens (M) | $\Delta$ Acc vs SC (pp) | Savings vs SC (%) |
> |:--|--:|:--|--:|--:|--:|--:|
> | **HMMT** | **128** | SC (Self-Consistency) | 62.45 | 69.11 | 0.00 | 0.0 |
> | **HMMT** | **128** | Semantic-Entropy-0.8 | 62.34 | 51.22 | -0.10 | 25.9 |
> | **HMMT** | **128** | Semantic-Entropy-0.5 | 62.34 | 56.03 | -0.10 | 18.9 |
> | **HMMT** | **128** | **DeepConf (ours)** | **62.66** | **43.37** | **+0.21** | **37.2** |
> | **HMMT** | **512** | SC (Self-Consistency) | 63.33 | 276.40 | 0.00 | 0.0 |
> | **HMMT** | **512** | Semantic-Entropy-0.8 | 63.33 | 195.17 | 0.00 | 29.4 |
> | **HMMT** | **512** | Semantic-Entropy-0.5 | 63.33 | 217.38 | 0.00 | 21.4 |
> | **HMMT** | **512** | **DeepConf (ours)** | **64.48** | **155.41** | **+1.15** | **43.8** |
>
> Regarding self-certainty, our paper's mean-confidence baseline can be viewed as a direct instantiation of self-certainty: it aggregates per-token confidence over a completed reasoning trace to score each path. In offline settings, we find that our tail-confidence variant is as good as or stronger than mean-confidence, because the tail better reflects whether the reasoning arrives at a reliable conclusion. More importantly, mean-confidence/self-certainty is not applicable to online early-exit: it requires decoding an entire trace before computing the certainty score, so it cannot decide to stop mid-generation and therefore cannot save tokens in serving. In contrast, our local confidence (i.e., bottom 10% confidence) is computed on-the-fly during generation, enabling online pruning and early exit that reduces token usage while improving accuracy.
>
> **Q2&W1:** Can the authors provide quantitative measures (e.g., AUROC, ECE) showing how well their confidence metrics discriminate correct vs. incorrect traces?
>
> We have performed a comprehensive analysis of AUROC (to measure discrimination) and ECE (to measure calibration) on the AIME24 and HMMT Feb 2025 datasets using Qwen3-32B. We evaluated three variants of our metric: Mean Confidence, Bottom 10% (Sliding Window), and Tail Confidence.
> As shown in the table below, all three metrics demonstrate strong discriminative power, consistently achieving high AUROC scores (>0.90) across both datasets. This indicates that our proposed local confidence signals (Bottom 10% and Tail) are highly effective at distinguishing correct traces from incorrect ones.
> Regarding calibration, the metrics are also highly reliable. After applying standard Platt Scaling (Logistic Regression) to map the scores to probabilities, we achieve an ECE of <0.05 across all variants. This quantitative evidence confirms that the model's internal confidence estimates are reliable and that "confidently wrong" errors are rare enough not to disrupt the ranking order.
>
> Table 1: Quantitative Evaluation of Confidence Metrics (AUROC & ECE)
>
>  | Dataset | Metric | AUROC ($\uparrow$) | ECE (Platt) ($\downarrow$) |
> | :-- | :-- | :-- | :-- |
> | AIME24 | Mean Confidence | 0.9151 | 0.0371 |
> | | Bottom 0.1  | 0.9088 | 0.0253 |
> | | Tail 2048 | 0.9006 | 0.0442 |
> | HMMT (Feb 25) | Mean Confidence | 0.9220 | 0.0346 |
>  | | Bottom 0.1  | 0.9148 | 0.0300 |
>  | | Tail 2048 | 0.9052 | 0.0195 |
>
> Note: ECE is reported after 5-fold cross-validated Platt scaling. This standard step is necessary to map our raw, non-probabilistic confidence scores into probabilities for a valid calibration analysis.

---

> > ### Author Response · Authors · 2025-11-25
> >
> > **Q3&W2:** How often do "confidently wrong" traces dominate the vote, and can local diversity or disagreement be incorporated as a second signal?
> >
> > Statistically, "confidently wrong" traces are rare outliers, not the norm. Our analysis shows that: on AIME24 and HMMT, our confidence metrics consistently achieve an AUROC over 0.90. This high score confirms that in the vast majority of cases, the model reliably assigns lower confidence to incorrect answers.
> > However, outliers do occur. For example, on AIME 2025 (Question 30), we found an incorrect trace from Qwen3-8B with a confidence score of 23.09, which overshadowed a correct trace scoring 14.35. While these cases are infrequent, they are real. We agree that incorporating diversity is a promising mitigation. DeepConf is fully compatible with such signals; future work can combine our local confidence metrics with a diversity-based re-weighting (e.g., penalizing clusters of identical high-confidence errors) to robustly handle these edge cases.
> >
> > **Q4:** Can the same percentile threshold s learned during warm-up be reused across multiple problems from the same domain, or must they be recomputed for every new query?
> >
> > We find that fixed thresholds are not transferable. Confidence distributions vary too much with problem difficulty. For example, on all questions of AIME 2025, the top-10% threshold for Qwen3-8B ranged from 11.5 to 15.5. A single static number fails here. It ends up being too loose for easy questions and too strict for hard ones. Therefore, filtering must be query-adaptive, adjusting dynamically to the current input rather than relying on a fixed global value, even for problems from the same domain.
> >
> > **Q5&W3:** Hyperparameters: The choice of top-k = 20 and window = 2048 seems arbitrary. An additional ablation varying these values could offer deeper understanding of sensitivity. Would varying the top-k (for confidence) or window size meaningfully affect results?
> >
> > For Top-K = 20: This choice is empirically grounded in the probability sparsity of modern LLMs. Our analysis of randomly sampled traces shows that for >99.7% of token positions, the cumulative probability mass of the top-20 tokens exceeds 0.99. This confirms that K=20 captures virtually most relevant signal while minimizing computation. Practically, this also aligns with standard serving engines like vLLM, which typically return a maximum of 20 log-probabilities.
> >
> > For the window size=2048: We chose 2048 as our default window size. As Table 8 shows, performance remains stable for sizes between 512 and 2048. We picked 2048 because it balances sensitivity and robustness. It is sensitive enough to catch real reasoning errors, yet stable enough to ignore minor noise. This setting worked consistently across all the models and datasets we tested.

---

### Meta-Review · Area_Chair_joVc · 2025-12-19

**Summary:**

The initial concerns primarily involved detailed experimental settings and results. Most of these have been adequately addressed in the authors' rebuttal. However, two points raised by one reviewer remain outstanding: 1. The potential for the LLM confidence measure to be incorrect. 2. Questionable generalizability to open-ended tasks.

Regarding the first point, the authors acknowledge the limitation and provide empirical studies to demonstrate that their proposed confidence criterion is reasonable, though it is not guaranteed. For the second, they clarify that their work is not focused on open-ended tasks—a fair delimitation of scope.

Given that the core concerns have been addressed and the remaining points are acknowledged limitations or scope clarifications, I believe this is a borderline but acceptable paper worthy of publication at ICLR.

**Reviewer Concerns:**

As summarized in the Summary section, the initial concerns primarily involved detailed experimental settings and results. Most of these have been adequately addressed in the authors' rebuttal. However, two points raised by one reviewer remain outstanding: 1. The potential for the LLM confidence measure to be incorrect. 2. Questionable generalizability to open-ended tasks.

**Reviewer Scores:**

Reviewer yu1z has stated after the authors' rebuttal that (s)he will increase her/his rating from 4 to 6. For all the other 3 reviewers, I think they would maintain their ratings.

---

### Decision · Program_Chairs · 2026-01-26

Accept (Poster)